# HERVs and Cancer—A Comprehensive Review of the Relationship of Human Endogenous Retroviruses and Human Cancers

**DOI:** 10.3390/biomedicines11030936

**Published:** 2023-03-17

**Authors:** Erik Stricker, Erin C. Peckham-Gregory, Michael E. Scheurer

**Affiliations:** 1Department of Molecular Virology and Microbiology, Baylor College of Medicine, Houston, TX 77047, USA; 2Department of Pediatrics, Baylor College of Medicine, Houston, TX 77047, USA

**Keywords:** human endogenous retrovirus, breast cancer, leukemia, lymphoma, skin cancer, reproductive cancer, liver cancer, prostate cancer, gastrointestinal cancer, renal cancer

## Abstract

Genomic instability and genetic mutations can lead to exhibition of several cancer hallmarks in affected cells such as sustained proliferative signaling, evasion of growth suppression, activated invasion, deregulation of cellular energetics, and avoidance of immune destruction. Similar biological changes have been observed to be a result of pathogenic viruses and, in some cases, have been linked to virus-induced cancers. Human endogenous retroviruses (HERVs), once external pathogens, now occupy more than 8% of the human genome, representing the merge of genomic and external factors. In this review, we outline all reported effects of HERVs on cancer development and discuss the HERV targets most suitable for cancer treatments as well as ongoing clinical trials for HERV-targeting drugs. We reviewed all currently available reports of the effects of HERVs on human cancers including solid tumors, lymphomas, and leukemias. Our review highlights the central roles of HERV genes, such as *gag*, *env*, *pol*, *np9*, and *rec* in immune regulation, checkpoint blockade, cell differentiation, cell fusion, proliferation, metastasis, and cell transformation. In addition, we summarize the involvement of HERV long terminal repeat (LTR) regions in transcriptional regulation, creation of fusion proteins, expression of long non-coding RNAs (lncRNAs), and promotion of genome instability through recombination.

## 1. Introduction

As the first treatment targeting a human endogenous retrovirus (HERV) protein enters into phase III clinical trials to combat multiple sclerosis (MS) progression [1], HERVs are becoming increasingly promising targets in cancer diagnostics and therapies. Previously, HERVs have been identified as involved in the progression of complex diseases such as MS [1,2], schizophrenia [3,4], and type 1 diabetes [5,6]. Lessons learned from these phenotypes have illuminated the potential for HERVs to impact human cancer development and treatment options. For example, there is currently a phase I trial testing the safety of a HERV-E-derived peptide autologous T-cell (HERV-E TCR T-cell) therapy to treat clear cell renal cell carcinoma [7]. HERVs are single-stranded enveloped RNA viruses that integrated into the human germline by means of their long terminal repeats (LTRs) [8]. Astonishingly, over 8% of the human genome comprise HERV sequences [9,10]. The LTRs in simple retroviruses, such as HERVs of the gammaretroviral subfamily, flank the capsid (*gag*), polymerase and protease (*pol*), and envelope (*env*) genes. HERVs of the betaretroviral (e.g., HERV-K) or spumaviral (e.g., HERV-L) subfamily carry additional non-structural genes such as HERV-K *rec*, HERV-K *np9* and HERV-L *tas*/*bel1*, and HERV-L *bet*, respectively [11]. In contrast to infectious retroviruses, about 87% of HERV sequences in the human genome are remnants of proviruses embodied by solo LTRs, while about 1.5% and 11.5% of HERV loci carry complete and truncated genomes, respectively [12]. For this reason, HERVs were historically considered “junk” DNA [13]. However, discoveries from the last two decades revealed substantial functions for various HERV elements in immune regulation [14,15,16], cell differentiation [17,18], cell fusion [19,20], transcriptional regulation [21,22,23,24,25], and cell transformation [26,27]. With this increasing evidence on the role of HERVs in cancer development, summaries of the current scientific knowledge are of high importance.

Even though similar endogenous retroviruses can be found in a variety of mammal species, articles on animal endogenous retroviruses (ERVs) were not considered for this review as it has been shown that data from retroviral animal systems have limited applicability to humans [28]. While insertions and oncogene activation through ERVs has been observed in mice [29] and chickens [30,31], leading to the hypothesis of HERV retrotransposition as a cancer driver [32,33,34], no de novo germline or somatic insertions of HERVs could be identified despite the sequencing of more than 4000 human cancer genomes over the last 25 years [34]. In the search of actively retrotransposing elements in humans, non-LTR LINE elements have been identified as better substrates for non-allelic recombination [35,36,37,38] Their central role in pathogenic recombination is further supported by their 5 times higher abundance in the human genomes than HERV sequences and the observation that reverse transcriptase (RT) inhibitors display effects as treatment in some human cancers [39]. Nonetheless, HERV insertion points have been reported in several cancer-related pathway genes, which utilize their viral proteins and altering transcriptional control mechanisms to initiate or promote oncogenesis.

Previous reviews of HERVs and cancer development have focused on either a specific gene product or functional region (e.g., Env [20,40,41], LTR [42], HERV-LTR7 lincROR (HGCN: LINC01419) [43], a particular family (e.g., HERV-W [44,45], HERV-E [46], HERV-K [47]), or a certain role of HERVs (e.g., transcriptional regulation [48], promoter exaptation [49], chromosomal rearrangement [50]). While uncovering specific aspects of HERV biology, these articles often fail to capture the interplay of different molecular mechanisms from multiple HERV families on specific cancer types. Other review articles categorize HERV effects on cancer development by molecular mechanism [51,52,53,54] and accordingly use only examples from different cancers providing no overview of the network of HERVs acting in a specific cancer type. Additionally, general review articles on HERVs and cancer tend to focus on the most common and clinically prevalent cancers, while articles on specific cancers are only available for a small number of main cancer types (e.g., breast cancer [55], melanoma [56], Hodgkin’s lymphoma [57], and prostate cancer [58]).

Therefore, we conducted a review on the carcinogenic impacts of HERVs by cancer type. While the literature on HERVs is growing exponentially due to improved sequencing and bioinformatic detection methods, it becomes increasingly challenging to provide a thorough overview of the cancers associated with HERVs. Fortunately, recent advances in the digital availability of scientific articles and improvements in search algorithms have provided a way to filter and evaluate large numbers of published articles. Consequently, systematic reviews are increasing in number, making complex topics available to clinical, translational, and basic science researchers in a digestible way. With their recent publication, Grabski et al. (2019) [59] were the first to conduct a systematic review employing a search algorithm for the evaluation of clinically relevant links between HERVs and cancer. While Grabski et al. provide an overview of the HERV-facilitated treatment and diagnostic options of surgical diseases, we set out to conduct a review of the molecular pathways and mechanisms affected by HERVs across the spectrum of cancer types. Our goal was to feature the integration of HERVs in known cancer pathways and to outline HERVs that are potentially suitable for the development of targeted cancer therapies.

Certain HERVs have perfectly assimilated into the cellular environment preventing oncogenesis, while others maintain their pathogenic potential and remain undercover until a time of cellular dysregulation. Accordingly, the goal of this review is to provide a comprehensive overview of the HERV influences for the human cancers listed below. Greek letters in superscript were used to denote viral functional regions or genes associated with specific observations (see Table 1).

## 2. Methodology Used to Obtain Primary Cancer-Related HERV Literature for a Qualitative Review

To allow for thorough coverage of the primary literature for this umbrella review, we identified articles on HERVs published through PubMed using the following search terms in titles and abstracts: ((HERV [Title/Abstract]) OR (ERV [Title/Abstract]) OR (endogenous retrovirus [Title/Abstract]) OR (endogenous retroviral [Title/Abstract])). Using the corresponding filter in PubMed, we identified review articles and evaluated them for articles missed by the PubMed search. We downloaded open access papers using the PubMed-Batch-Download software developed by Bill Greenwald [60], supplemented with a manual download through PubMed with Texas Medical Center (TMC) library access in portable document format (PDF). We obtained articles with restricted access through the Texas Medical Center Library using the OpenAthens plugin in EndNote. Only articles accessible and available in English were assessed. We used our recently developed R package called PDF data extractor (PDE) available on CRAN (https://CRAN.R-project.org/package=PDE, version 1.4.3 accessed on 6 February 2023, Houston, TX, USA) as a pre-screening tool to separate cancer- from non-cancer-related articles [61]. We manually divided the articles by cancer type and later grouped them for this review. Lastly, we evaluated the identified full-text articles ad hoc for cancer-specific factors inducing HERV expression; HERV insertions, deletions, or recombinations distinctively observed in a cancer type; interactions of HERVs with known oncogenes, tumor suppressors, or other signaling pathways; HERV products with functional effects in a cancer; and HERV-based treatment approaches. The chapters in this review were similarly structured to provide a coherent hierarchy for the reader. For the qualitative synthesis, we weighted the scientific findings by number of articles reporting similar results, and excluded findings that were based on inaccurately described or performed methods (e.g., lack of detail, no technical replicates, inappropriate method for a certain conclusion). Articles on animal endogenous retroviruses were not considered for this study as it has been shown that data from animal systems are minimally applicable to human systems [28]. In the same way, papers on non-human cancers were included only at a minimal level in this review. Graphical summary figures were created with BioRender (accessed on 3 February 2023, Houston, TX, USA).

## 3. HERVs in Breast Cancer—The Rise of New Biomarkers

Breast cancer is the most common cancer and the leading cause of cancer-related deaths in women worldwide [62]. Due to the many breast cancer subtypes and their varying treatment responses [63], targeted treatments that evolved in recent years have become a success story. However, the field is still in need of preventive and early detection methods.

HERVs might be able to close this gap providing new targets for prognostics, diagnostics, and treatments. Several groups have independently reported the overexpression of messenger RNAs (mRNAs) and proteins from multiple HERV families in breast cancer cell lines and patient tissues compared to healthy tissues [64]. Interestingly, the menstruation-associated hormones estradiol and progesterone were observed to increase HERV-K (HML-4) *env* [65] and HERV-K (HML-4) RT transcripts as well as HERV-K (HML-4) RT protein levels [66] in breast cancer cell lines (Figure 1). In breast cancer patients, increased HERV-K (HML-4) RT as well as HERV-K (HML-4) Env protein levels were shown to be associated with shorter metastasis-free and overall survival [66,67]. Conversely, Montesion et al. (2018) [68] identified two HERV-K (HML-2) LTRs (HGCN: *ERVK-5* at position 3q12.3 and *ERVK3-4* at 11p15.4) that had specifically increased promoter activity in breast cancer while decreased activity in immortalized human mammary epithelial cells. Additionally, several stage-specific transcription factor (TF)-binding sites within the two LTRs were predicted to potentially contribute to promoter activity during neoplasia [68]. While the *ERVK-5* (HERV-KII) was fixed in humans, the *ERVK3-4* (HERV-K7) was found to be polymorphic in the human population with an allele frequency of 51%, presenting the prospect of a newly identified risk facto r [68]. In addition, breast cancer cell lines were shown to harbor HERV-K111 gene conversion/deletion events in the pericentromeric region of chromosome 22, suggesting a contribution to genomic instability [69]. Furthermore, certain HERV-K (HML-2) Env splice variants have been suggested as breast cancer-specific antigens but are still under investigation [65].

Higher expression of particular HERV-K gene products in patients with breast cancer is also signified by the pronounced immune response against such proteins. Responses include increased T-cell proliferation, Th1-specific cytokine secretion (i.e., INFγ, IL2, IL6, CXCL8, CXCL10), immune checkpoint activation, and serum antibody production against HERV-K (HML-2) proteins [70,71,72]. Additional to higher serum HERV-K mRNA levels and serum anti-HERV-K antibody titers in women with ductal carcinoma in situ and stage I disease compared to women without cancer, Wang-Johanning et al. (2014) reported that elevated HERV-K (HML-2) antibodies and mRNA levels in the blood can be an early indicator of future metastatic disease development [72]. In patients undergoing chemotherapy, HERV-H, -K, -R, and -P *env* mRNA expression is reported to be decreased compared to patients not receiving chemotherapy [73]. In mouse model systems, treatment with HERV-K (HML-2) Env-directed antibodies [74], short interfering RNA-mediated knockdown (RNAi) of HERV-K (HML-2) *env* [75], and chimeric antigen receptor (CAR) T cells specific for HERV-K (HML-2) Env protein [76] were able to recapitulate the effects against the engrafted human tumors. For all three treatments, specific cytotoxic effects against tumor cells were observed in the mice: reduced tumor growth, induced apoptosis, and reduced metastasis [74,76].

A detailed study of the HERV-K (HML-2) *env* knockdown through RNAi revealed the involvement of the viral gene in cellular pathways playing key roles in cancer (e.g., *EGFR*, *TGF-β*, *NF-κB*, *MYC*, *p53*, *HRAS*, *KRAS*, and *MAPK1/3*) (Figure 2) [75]. Overexpression of HERV-K (HML-2) *env*, on the other hand, increased breast cancer cell transformation, migration, and invasion, as well as restored the cancer-related signaling pathways mentioned above alongside the downregulation of p53 (HGNC: TP53) [75]. Additionally, microarrays identified HERV-K (HML-2) Env protein as a strong inducer of the MAPK pathway via upstream TFs [77], and examinations of the DNA methylome and TF-binding data revealed several HERV LTR77-driven TFs such as NF-κB (HGNC: NFKB1) and RAD21 [41,78]. Besides HERV-K (HML-2) Env, HERV-K (HML-2) non-structural nuclear protein Np9 has been described to interact with cellular proteins [79]. Np9 destabilizes LNX1, an E3 ubiquitin ligase, that targets members of the NUMB/NOTCH1 pathway for degradation [79,80]. NOTCH1 regulates cell differentiation, cellular metabolism, cell cycle progression, angiogenesis, self-renewal, and immune function [81] and has been shown to be deregulated in breast cancer [82]. Another ubiquitin ligase directly bound by Np9, which has been found upregulated in breast and other cancers, is MDM2 [27]. Contrary to the inhibition of p53 observed for HERV-K (HML-2) Env, Np9 has been reported to interfere with the MDM2 ubiquitin ligase activity toward p53 in the cell nucleus and thus increases p53 levels while being ubiquitinated itself [27]. Furthermore, Np9 was discovered to result in an upregulation of CD147 (HGNC: BSG) [83]. CD147 is a coreceptor for VEGFR2 (HGNC: KDR) and has been demonstrated to induce VEGFA [84,85] in addition to ADAMTS1 and ADAMTS9 [86]. Both VEGFA/VEGFR2 and ADAMTS1/9 signaling are reported inducers of metastasis and angiogenesis in several cancers [84].

Jin et al. (2019) [87] uncovered an HERV-derived long noncoding RNA (lncRNA), named TROJAN^λ^, which is highly expressed in human triple-negative breast cancer (TNBC). This LTR70-driven lncRNA is a promising therapeutic target, as it binds ZMYND8, a metastasis-repressing factor, and leads to its degradation by the ubiquitin-proteasome pathway [87]. Whereas TROJAN^λ^ overexpression stimulated TNBC proliferation and metastasis in vitro, in vivo studies in mice confirmed that RNAi targeting TROJAN can inhibit TNBC progression and reduce tumor size [87]. Similarly, several groups have proposed treatment strategies targeting HERV proteins as tumor-specific antigens [88,89]. The potential success of such approaches is supported by the findings of Sheng et al. (2018) who showed that the genetic or pharmacological ablation of the histone demethylase LSD1 (HGNC: KDM1A) enhances tumor immunogenicity by stimulating HERV expression [90]. In addition to enabling HERV transcription, KDM1A elimination was observed to prevent the removal of an inhibitory methyl mark on the RNA-induced silencing complex (RISC), which in its usual function promotes the degradation of HERV mRNAs [90].

## 4. HERVs in Lymphoma—The Silent Inducers

Lymphomas are characterized by an increased proliferation of lymphocytes and are classified according to their maturity (peripheral or mature versus precursor) and cell lineage (B, T, or natural killer cell) [91]. In contrast to other cancers, a hallmark of lymphoma is its origin in the immune system, with known risk factors that perturb immune functions, such as immunosuppressive drugs [92], autoimmune disorders [93], and viral infections including Epstein–Barr virus (EBV), human immunodeficiency virus (HIV), or human T-cell lymphotropic virus (HTLV) [94]. As there is complex interaction between HERVs and immune system function, it is consistent with previous literature that HERV deregulation influences lymphomagenesis.

HERV-K (HML-2) was shown to have markedly different titers in the blood of patients with lymphoma (e.g., HIV infection with diffuse large B-cell lymphoma (DLBCL), non-HIV diffuse large B-cell lymphoma, and HIV infection with Hodgkin lymphoma (HL)) compared to healthy individuals [95]. Remission of the cancer after successful treatment was associated with a significant decrease in viral titers [95]. While there was a large range of titer differences between patients with lymphoma (on average 10^10^ copies/mL) and healthy individuals (on average 10^2^ copies/mL), HERV-K viral particles were found in the plasma of all patients with lymphoma [95]. Additionally, the differences in titer between lymphoma groups (e.g., high titers of HERV-K (HML-2) type 1 with low HERV-K (HML-2) type 2 titers in HL) suggests the use of specific HERV-K titers as sensitive and specific biomarkers for lymphoma [95]. Furthermore, immune histochemical staining showed that cutaneous T-cell lymphoma (CTCL)-derived extracellular vesicles were positive for syncytin-1 (HGCN: ERVW-1^ε^) [96].

While HERV-K (HML-2) was not found to be induced by HTLV [97], Leung et al. (2018) reported that EBV required the activation of HERV LTRs for transcription of oncogenic genes such as *HUWE1* (Figure 3A) [98]. In 2015, Zahn et al. identified HERV-K111 as an additional putative biomarker for lymphoma while studying the pericentromeric regions of chromosomes [99]. They observed the absence of the pericentromeric HERV-K111 5′ regions in cutaneous T-cell lymphoma (CTCL) lines HUT78 and H9 and mutated HERV-K111 in Jurkat cells [99]; however, Kaplan et al. (2019) confirmed a significant increase in the homozygous absence of the HERV-K111 5′ regions in non-Hispanic White patients with severe CTCL compared to healthy individuals [100]. As HERV-K111 exists in over 1000 copies in the pericentromeric region, a deletion of about 3400 Kb is suggested [100]. Thus, pericentromeric instability is a likely consequence of the missing sections but will have to be confirmed by additional experiments.

There are only two studies reporting antibodies against HERVs in 2–6% of patients with lymphoma (N = 288) [101,102]. This could be due to reduced immune function in these patients or the very nature of HERV integration in lymphoma development. Transcriptional activation of normally dormant oncogenes by HERV LTRs, so called promoter exaptation, appears to play a much larger role during lymphocyte transformation than the interference in cancer pathways by HERV proteins or gene products, thus providing no targets for the immune system [49]. The first ever described case of retroviral promoter exaptation was the expression of the CSF1 receptor (*CSF1R*) in HL cells discovered by Lamprecht et al. (2010) [103]. Autocrine stimulation by the macrophage growth factor CSF1 has been shown to be essential for the proliferation and survival of HL cells, and lineage inappropriate expression of the *CSF1R* has become a hallmark in HL [103]. In contrast to regular macrophages, CSF1R in HL cells is driven by the THE1B LTR located 6.2 kb upstream of the locus and has been linked to loss of transcriptional repressor *CBFA2T3* expression [103]. In a similar way, normally brain-expressed fatty acid-binding protein 7 (*FABP7*) was observed to be expressed as a chimeric isoform with LTR2 in tissues from patients with DLBCL [104]. RNAi-mediated knockdown of *FABP7* resulted in decreased proliferation and growth of DLBCL, suggesting a dependence on *FABP7* expression [104]. A third example of promoter exaptation is the upregulation of IRF5 driven by the demethylated LOR1a LTR element that was specifically detected in HL cell lines [105]. IRF5 is a key regulator of the aberrant transcriptome in HL and crucial for HL cell survival [106].

Contrary to the pathogenic effects of HERVs described above, the double-copy HERV-R on chromosomes 7q11.21 and 7q33 (HERV-R.3-1 and HERV-R.3-2, respectively) has been classified to have tumor suppressive functions [107]. HERV-R.3-1 Env (HGNC: ERV3-1^ε^) was observed to be downregulated in HL cells compared to normal blood cells [107], which parallels the absence of expression seen in choriocarcinoma [108]. In choriocarcinoma cells, *ERV3-1* overexpression inhibits cell proliferation [109], while ERV3-1 is upregulated during terminal differentiation of leukemia cells and is highest in cell cycle arrested cells [107,110,111]. Downregulation of cyclin B and upregulation of the cyclin-dependent kinase inhibitor P21 (HGNC: CDKN1A) are likely to be key mechanisms for growth inhibition [109]. Interestingly, 1% of healthy non-Hispanic Whites carry a stop codon in the *ERV3-1* region in a homozygous state without a pathogenic phenotype, questioning the impact of HERV-R.3-1 on at least essential physiological functions [112]. Another interesting finding in HL cells is the interaction of a dual specificity phosphatase 5 (*DUSP5*) pseudogene and a specific HERV sequence. While most HERV families can be found in different species of mammals, HERVs of the family K are quite unique to the human genome [113]. Therefore, it was striking to find the pseudogene of *DUSP5* (*DUSP5P1*), inserted in the HERV-K_1q42.13 sequence, indicating the prior presence of the HERV-K family in that location [113]. Most striking was the observation that several cancer cell lines (Burkitt’s lymphoma, leukemia, neuroblastoma, and Ewing sarcoma) as well as peripheral blood mononuclear cells (PBMC) from patients with HL displayed a significantly higher ratio of *DUSP5P1*/*DUSP5* expression than normal cells [113]. While the DUSP5/DUSP5P1 ratio correlated with levels of the pro-apoptotic factor B-cell leukemia/lymphoma 2-like 11 (BCL2L11), the authors hypothesized that DUSP5P1 reduces the activity of DUSP5 via RNAi or competitive inhibition results in increased MAPK1/3 activity and subsequent inhibition of BCL2L11 [113]. However, a distinct function for DUSP5P1 is yet to be determined [113].

Besides cancer pathways in which HERVs have incorporated themselves, advances in treatments have revealed HERVs being part of therapy-related side effects and drug resistance mechanisms. ABCB1 (MDR-1) is one of the most expansively studied drug resistance mechanisms [114]. The *ABCB1* gene encodes a 170-kDa ATP-dependent efflux pump for the plasma membrane, which prevents intracellular drug accumulation [114]. Interestingly, aberrant MDR-1 transcription found in lymphoma cells is driven by the ERV1 LTR MER57 initiating transcription in the opposite direction of the viral promoter [114]. To reduce the transcription of detrimental genes, several DNA methyltransferases (DNMT) and histone deacetylase (HDAC) inhibitors have been incorporated in the treatment plan for hematopoietic and lymphatic malignancies in recent years [115]. Currently, it is not fully understood if DNMT and HDAC inhibitors provide any beneficial effects through the activation of beneficial HERVs or if the inhibitors lead to an interference with drug-induced HERV elements. Studies indicated that especially LTR12C elements from the HERV9 family are activated by DNMT and HDAC inhibitors [115,116] and further stimulated when vitamin C is also taken [117]. Even though the fear of aberrant expression of oncogenes and HERVs with unknown function is large, several LTR12 elements have been detected to induce tumor suppressor genes (see HERVs in Testicular Cancer—The Governors of Tumor Suppressor Genes) [118,119,120]. Additionally, HDAC inhibitors have been shown to prevent the activation of HERV-L, which might participate in pathogenesis [116]. New research is required to evaluate the HERV-related side effects of such drugs.

## 5. HERVs in Leukemia—The Lifesavers for Cancer Cells

Leukemias are the most common childhood cancers worldwide [62,121] and among the cancers with the lowest somatic mutational burden [122]. Both characteristics suggest genomic risk factors that can be inherited, and when accumulated, lead to carcinogenesis. Analogous to observations made for HERV LTRs in prostate cancer (see HERVs in Prostate Cancer—The Dancing Partner of the Androgen Receptor), THE-7 LTRs were discovered as drivers of a translocation of chromosome 14q32 to chromosome 7q21 in a female patient with B-cell chronic lymphocytic leukemia (B-CLL) (Figure 3B) [123]. Furthermore, fibroblast growth factor receptor 1 (*FGFR1*) was found to be constitutively activated through the fusion between a HERV-K3 (HML-6) sequence (HGCN: ERVK3-1) and the *FGFR1* gene in a male patient with an atypical stem cell myeloproliferative disorder [124,125]. The fusion and resulting aberrant growth signal were the result of a translocation involving chromosomes 19q13.3 and chromosome 8q12 [124,125]. Furthermore, deletions of pericentromeric HERV-K111 regions in adult T-cell leukemia cell lines were enriched leading to chromosomal instabilities [69]. Additional to these chromosomal abnormalities as prognostic markers, Schmidt et al., (2015) identified single nucleotide polymorphism (SNP) markers near two endogenous retroviral loci, HERV-K (HML-2) on chromosome 1 (HGCN: *ERVK-7*) and HERV-Fc1 on chromosome X (HGCN: *ERVFC1*) associated with multiple myeloma [126]. Both HERV regions encode nearly complete viral proteins, suggesting a functional involvement of the gene products in disease development [126].

Similar to the observations in lymphomas, the immune response against HERV-K (HML-2) and other HERVs appears to be rather weak or unexplored (see also HERVs in Lymphoma—The Silent Inducers) [100]. This might also be due to the fact that HERV-K108 (HGCN: ERVK-6) Env TM has immunosuppressive properties and has been reported to induce IL10 in PBMCs [127]. IL10 is an anti-inflammatory cytokine, which terminates T-cell responses and leads to immune tolerance [128]. In a comparable way, surface *CD5* expression on B cells regulates their functional fate and immunological activity. *CD5* expression is tightly controlled through a HERV-E sequence located upstream of the *CD5* locus (*HERV-E::CD5*). The *HERV-E::CD5* sequence was shown to induce the integration of an alternate exon, resulting in low levels of membrane CD5 in normal B cells [129]. Conversely, high levels of *CD5* expression in B cells caused by the absence of the alternate exon was found to be associated with CLL [129].

Another oncogenic function of HERVs involves *np9*, which is among the HERV-K genes overexpressed in leukemias (Figure 2) [79,130,131]. Np9 was shown to not only activate MAPK, AKT, and NOTCH1 signaling pathways (see Results HERVs in Breast Cancer—The Rise of New Biomarkers), but also to upregulate β-catenin, which is (HGNC: CTNNB1) essential for survival of leukemia stem cells [26,132]. Silencing of *np9* in turn inhibited the growth of myeloid and lymphoblastic leukemia cells [26]. In several systematic studies, Sokol et al. uncovered various physiological transcription regulation mechanisms of HERV sequences and showed that HERV9 LTRs (LTR12) control splicing of the tumor suppressor genes *CADM2* and SEMA3A in erythroleukemia and human embryonic stem cells [133,134]. Additional beneficial HERV9 functions are discussed in HERVs in Testicular Cancer—The Governors of Tumor Suppressor Genes. In addition, the HERV-H/F locus on chromosome 6 was described to carry several TF-binding sites involved in normal hematopoiesis and with reduced expression in B-cell and myeloid lineage leukemia, indicating its transcription under normal conditions [135].

Further evidence for physiological functions of HERV gene products is the observation that 5-azacitidine, a DNA demethylating agent used to treat preleukemia and leukemias, activates cancer/testis antigens (CTAs) [136,137,138] and HERV gene products specifically in tumors, triggering innate immunity [139,140]. This phenomenon was also observed in several other cancers including colorectal [140], urothelial [141], and melanoma [139,142].

## 6. HERVs in Skin Cancer—The Highly Addictive Treatment Targets

Compared to other organs, the skin is exposed to some of the highest amounts of mutagens; therefore, skin cancer is the malignancy with the highest mutational burden [122]. Accordingly, several physical and chemical agents with mutagenic potential have been proven to influence the regulation of HERV sequences [9,10]. As such, UV radiation, the primary risk factor for both melanomas and non-epithelial skin cancers, was shown to induce *gag* expression of HERV-K (HML-2) [143] in melanoma cell lines and tumor tissues; to reduce *rec* and *np9* expression of HERV-K (HML-2) in primary human melanocytes and melanoma [144]; and to reduce *pol* expression of HERV-K, -H, -L, -FRD, -E, and ERV9 in melanoma cell lines [145], primary keratinocytes [146], and skin biopsies (Figure 4) [147]. Furthermore, Karimi et al. (2018) demonstrated that copper (a potent antibacterial agent and deregulated nutrient in cancer) in the form of CuSO_4_ increases HERV-K (HML-2) and -W *env* transcripts in the melanoma cell line SK-Mel-37 [148]. HERV-K (HML-2) gene expression as a hallmark of melanoma was confirmed by multiple groups and was reported to be further induced in melanoma cells by serum starvation [149,150].

Besides HERVs, many oncogenes have been found in higher levels in skin cancers, likely caused by mutated TF-binding sites in promoters [151]. HERV LTRs have been postulated to harbor over 64% of all human-specific TF-binding sites in human embryonic stem cells [152], and chromatin immunoprecipitation (ChIP) assays as well as gene expression studies revealed that one third of p53 (HGCN: TP53) sites are located within HERV LTRs [153]. Of all LTR-associated p53-binding sites, approximately 70–90% are located in sequences of the ERV1 family [154], while HERV-I LTRs have been shown to be repressed by TP53 and activated by TP53 mutations [155]. Additional to many more TF-binding sites located in HERV sequences (a web-based browser developed by Garazha et al. (2015) [156] is available at https://herv.pparser.net/GenomeBrowser.php (accessed on 6 February 2023)), Sp1, Sp3, and YY1 have been described to specifically induce unmethylated HERV-K (HML-2) LTRs in melanoma cell lines [157]. Particularly, hypomethylation of HERV-K6 at 7q22.1 (HCGN: ERVK-6) and LINE-1 elements was observed to be associated with worse prognosis and poorer survival for melanoma patients [158].

Despite the general assumption that the majority of HERV sequences in the genome are inactive due to DNA methylation and other histone marks, a study published by Jacques et al. (2013) indicated that up to 80% of HERV regions exist in an open chromatin state [159]. Particularly in patients with melanoma, both lncRNAs, *BANCR* and *SAMMSON*, are promoted by HERV LTRs [49,160,161]. Both lncRNAs were confirmed to increase growth and invasiveness of melanocytes [160,161]. Conversely, *BANCR* knockdown was shown to reduce melanoma cell migration, which could be rescued by recombinant overexpression of *CXCL11*, indicating an inhibition of CXCL11 by *BANCR* [160]. *SAMMSON* was demonstrated to play an equally important role in melanoma formation, as it can be detected in more than 90% of human melanomas, functions as a lineage addiction oncogene [49,161] and is generally included in gene amplifications involving melanoma-specific oncogene *MITF* due to its proximity to the gene. MITF in turn has been reported to induce HERV-K (HML-2) gene expression by binding to E-box (CA(C/T)GTG) located within the HERV-K (HML-2) LTR [162,163]. *SAMMSON* directly binds p32, leading to its nuclear targeting and preventing its function in maintaining mitochondrial integrity and homeostasis [163].

Moreover, HERV-K (HML-2) expression was shown to be induced by the activation of the MAPK and CDKN2A-CDK4 pathways in melanoma cells [164]. In addition to statistically significant differences noted in the seroprevalence of HERV-K (HML-2) antibodies between patients with melanoma and healthy individuals, higher serological HERV-K (HML-2) reactivity was identified as an indicator of improved survival in patients with melanoma [165]. Increased expression of HERV-K (HML-6) in melanoma cell lines and tumor tissues has clinically been associated with higher HERV-K (HML-6) antibody levels [166]. While the contribution of yellow fever virus and BCG vaccine-associated HERV-K (HML-6) to the prevention of melanoma still remains a controversial topic [167,168], several studies report supporting evidence for the potential use of HERV-K (HML-2)-targeting strategies as treatment for melanomas. For instance, the use of RNAi-targeting HERV-K (HML-2) was shown to prevent the transition from an adherent to a non-adherent growth phenotype of melanoma cells [149], and CAR-T cells targeting HERV-K (HML-2) Env were observed to have a significant antitumor effect [169]. In addition to its structural functions, HERV-K (HML-2) Env also carries an immunosuppressive domain, which was shown to lead to the local immune evasion of B16 melanoma cells injected in mice [170]. Furthermore, HERV-K (HML-2) inhibition prevented the pathogenic formation of multinuclear atypia cells in melanoma [149]. Contrary to the observed reduced cell grow upon inhibition of *ERVW-1^ε^* in BeWo cancer cells [171], recombinant B16F10 melanoma cells with stable expression of *ERVW-1^ε^* displayed decreased cell proliferation, migration, and invasion [172]. Therefore, functions of *ERVW-1^ε^* in breast cancer cells might be cell type- and concentration-specific. Most noteworthy for possible targeted treatments is the discovery of an aggressive subpopulation of melanoma cells that are highly dependent on HERV-K (HML-2) activation [173]. In fact, patients with melanoma and autoimmune diseases display increased survival rates, suggesting that systemic autoimmunity also targets cancer-associated HERV activities [174]. One potential mechanistic explanation includes HERV-K (HML-2) *rec*. Its depletion was demonstrated to lead to upregulation of epithelial-to-mesenchymal-associated genes and an enhanced invasion phenotype of proliferative melanoma cells [162]. Additionally, *LSD1* depletion (see Results HERVs in Breast Cancer—The Rise of New Biomarkers for more details) in combination with anti-PD-1 antibody (HGNC: PDCD1) therapy was found to increase the immunogenicity and response to checkpoint blockade of refractory mouse melanomas, providing another way to improved treatment efficacies [90].

## 7. HERVs in Testicular Cancer—The Governors of Tumor Suppressor Genes

Although the secretion of HERV viral particles was first detected in placenta by Kalter et al. (1973) [175], three years later, testicular tumors became the first cancer tissue described to selectively release HERV viral particles [176]. Since then, testicular cancer cell lines serve as a model for HERV-K particle assembly and the effects of HERV-K protein expression on cell functions [177,178,179]. Cellular transcription factor YY1 was shown to specifically activate LTRs in teratocarcinoma cell lines [180] with the HERV-K (HML-2) LTR in Tera-1 cells becoming as strong of a promoter as the SV40 early promoter (Figure 5) [181]. Interestingly, several distinct testicular cancer cell lines and tissues were demonstrated to each specifically drive expression of reporter plasmids under different HERV-K (HML-2) LTR promoters [182,183]. Moreover, Mueller et al. (2018) revealed the capability of HERV-K (HML-2) to induce neighboring genes such as *PRODH* in germ cell tumors (GCT) that paralleled with the differentiation status of the cancer cells [184]. In this way, expression of genes adjacent to HERV-K (HML-2), which is highest in undifferentiated cells, may provide an additional indicator of the tumor status [184]. Conversely, transcription factor NFY induces HERV-9 LTR12-mediated transcription of tumor suppressors in normal testis, which is silenced in testicular cancer [119,120].

With HERV viral particles being secreted by testicular tumor cells, over 60% of males with germ cell cancers have been observed to display seroreactivity to HERV-K (HML-2) antigens, while only 4% of healthy individuals showed high antibody titers [177]. Additionally, HERV-K(HML-2)-specific T cells could be detected in patients with a history of testicular cancer, indicating immunological memory formation that may prevent relapse in certain cases [185]. Most significantly, HERV-K (HML-2) antibody seropositivity in patients with GCT that persisted after treatment was associated with both lower survival rates and lower chemotherapeutic success [186]. Thus, HERV-K(HML-2)-specific antibodies may have great diagnostic value as surrogate measurement for HERV-K (HML-2) expression and associated oncogenic effects in GCTs.

Specifically, the two viral proteins derived from alternative transcripts of the C-terminal portion of the HERV-K (HML-2) *env* gene were first discovered in GCT cell lines [187] and subsequently described to play a central role in the development of GCTs [188]. The proteins, later termed HERV-K (HML-2) Np9 [130] and Rec [189], associate in GCTs with the tumor suppressor PLZF (HGNC: ZBTB16), a transcriptional repressor, and chromatin remodeler and abolish its transcriptional repression of *MYC*, a major target of PLZF [190]. This leads to amplified cell proliferation and survival mediated by overexpression of *MYC* and corresponding MYC-regulated genes [190]. Furthermore, HERV-K (HML-2) *rec* expression in mice has been shown to result in the development of GCTs [191]. Np9, on the other hand, has been demonstrated to interact in addition to PLZF with multiple ubiquitin ligases modifying cancer pathways (see Results HERVs in Breast Cancer—The Rise of New Biomarkers ) [27,79,80]. The CRISPR-Cas9-mediated knockdown of Np9 was reported to increase sensitivity to bleomycin, cisplatin, and serum starvation and reduce migration, albeit not affecting the viability of NCCIT teratocarcinoma cells, which could be restored by adding recombinant Np9 [192]. Possibly less significant, but still noteworthy, is the detection of a HERV-H LTR/sPLA2L fusion protein, also called HHLA1^λ^, in teratocarcinoma cells lines [193]. While this fusion protein could not be confirmed in the patient samples tested [193], HHLA1^λ^ still has several functions in digestion and is postulated to contribute to chronic inflammation [194,195].

Current research reveals the pivotal role of LTR12-driven transcription in treatment of testicular cancer. LTR12 activity has been suggested as biomarker for the potency of anticancer drugs [119,120] and might be selectively stimulated for testicular cancer treatments. As indicated above, HERV9 LTR12 induces the transcription of tumor suppressors such as germ cell-associated, transcriptionally active *GTAp63* (HGNC: *TP63*) [118,119] and TNF Receptor Superfamily Member 10b (*TNFRSF10B*) [119,120]. Both tumor suppressors are frequently downregulated in testicular cancer cells and may be induced by HDACs to mediate apoptosis (see also HERVs in Lymphoma—The Silent Inducers) [119,120]. Moreover, lncRNAs driven by solitary HERV9 LTRs were found to function as SWI/SNF complex antagonist as well as targets of key TFs regulating proliferation, including NFY, TP53, and SP1 [196].

## 8. HERVs in Other Genital Cancers (Ovary Cancer, Choriocarcinoma, and Endometrial Cancer)—The Ascent of New Possibilities

Despite stably falling incidences of over the last three decades [197], most patients with ovarian cancer are not diagnosed until stages III (51%) and IV (29%) because they experience few or no symptoms until the disease has metastasized [198]. Therefore, HERVs have been investigated in different subtypes for their potential as prognostic, diagnostic, and therapy-resistance markers. Heidmann et al. (2017) proposed HEMO, a HERV MER34-derived Env protein, as a possible marker for ovarian clear-cell carcinoma (OCCC) as it was significantly increased in ovarian cancers with evidence for histiotype dependence [199]. In a similar way, HERV-W, HERV-E, and HERV-K (HML-2) displayed higher expression due to generalized hypomethylation in ovarian carcinomas compared to non-malignant ovarian tissues [200,201,202]. Intriguingly, hypomethylated HERV-K (HML-2) elements in OCCC were observed to be associated with poor prognosis, platinum-based therapy resistance, and increased metastasis (Figure 5) [201]. Contrary to this observation, Liu et al. (2018) reported methylation inhibitors and G9A inhibitors enhancing antitumor effects through viral mimicry [203]. Providing a potential explanation for this apparent discrepancy, the study investigators detected the most substantial effects of the inhibitors on HERV-Fc1 LTRs with only minimal effects on HERV-K (HML-2) MER9a1, highlighting potentially counteracting forces [203]. This can be further reconciled by the observation that, despite insufficient reactivity against the highly expressed HERV-K gene products, potent HERV-K-specific T cells can be generated from autologous dendritic cells (DCs) pulsed with HERV-K (HML-2) Env antigens. The T cells displayed high cytotoxicity against autologous tumor cells without affecting normal cells and could be further increased upon depletion of T regulatory cells [204]. A possible mechanism of HERV-Ks that counteracts the immune response and drives oncogenesis could include the ovarian cancer susceptibility gene *BRCA1* as it carries several HERV-K (HML-2) elements in its sequence [205]. As a marker for response to chemotherapy in ovarian cancer, a LTR-controlled alternate transcript of the molecular chaperone DNAJ has been suggested, as in its methylated form DNAJ is solely expressed in malignant and not surface ovarian cells [206].

ERVW-1^ε^ and ERVFRD-1^ε^ are the most extensively studied retroviral proteins encoded in the human genome and have a vital role in placentation by facilitating trophoblast fusion and creation of an immune privileged site [20,40]. Nonetheless, HERVs have been implicated in furthering abnormal growth and reduced differentiation of trophoblasts resulting in the development of choriocarcinoma. HERV-E induces expression of an alternative transcript of *PTN*, a heparin-binding protein with central functions in growth and differentiation control of the placenta [207,208,209]. The alternative *HERV-E:PTN*^λγπελθ^ transcript is driven by the TF SP1 binding to the HERV-E LTR in untranslated exon 1 of PTN [209]. *HERV-E:PTN*^λγπελθ^ mRNA was only detected in trophoblast cell cultures while absent in normal adult tissues (Figure 5) [208,209]. Interestingly, depletion of the *HERV-E:PTN*^λγπελθ^ transcript was shown to prevent human choriocarcinoma growth and invasion in a mouse model [208]. On the contrary, HERV-R.3-1 (HGCN: ERV3-1) has been proposed as tumor suppressor since its overexpression induced differentiation of the human BeWo choriocarcinoma cell line (see also HERVs in Lymphoma—The Silent Inducers) [109,210].

Various HERV *env* transcripts and proteins have been implicated in the development of endometrial cancer (e.g., HERV-E, Fc, FRD, H, K, R, Rb, T, V1, V2, and W *env*, as well as *ERVW-1* and *ERVFRD-1 RNA*) [211]. A demethylation-driven activation of the viral *ERVW-1* LTR through a functional estrogen receptor element (ERE) was suggested in 2012 [211]. The higher levels of ERVW-1^envW^ protein observed are theorized to then promote not only increased cell fusion, but also hyperproliferation, which is brought on by the interaction of ERVW-1^envW^ with tumor growth factor beta (TGF-β; HGNC: TGFB1) [211,212].

## 9. HERVs in Colorectal and Gastrointestinal Cancers—The Hopes and Hazards of Family H

Colorectal cancers (CRCs) are among the most common cancers worldwide with strikingly low 5-year survival rates of less than 65–70% in Northern America, Australia/New Zealand, and many European countries [62]. While early diagnosis has markedly improved for older patients due to routine screenings in individuals >50 years of age, rising rates in the population under 45 years of age highlight the need for improved non-invasive prognostic and diagnostic tools [213]. In the last decade, research has started to focus on the influence of HERV elements on CRCs and has revealed interesting interactions, especially involving HERV-H. With over 1000 loci in the human genome, HERV-H is the most abundant HERV family carrying coding regions in the human genome [214].

Findings by Liang et al. (2012) indicated that the number and composition of active HERV-H elements allow for the distinction between normal colon samples and colon tumor tissues, not necessarily the comparison of overall HERV H levels [188]. The study investigators observed 14 active elements in colon tumor tissues compared to 7 in adjacent normal tissues with some active loci located in close proximity to putative open reading frames, e.g., HERV-H_1q42.2 near viral RT and *env*, HERV-H_16q24.1 near viral RT, and HERV-H_19q13.31 near *CREB5* (Figure 6) [215]. Later, Pérot et al. (2015) detected a significant increase of HERV-H *gag*, *pol*, and *env* RNA in patients with CRC, with HERV-H loci on chromosome Xp22.3 (HGCN: ERVH-2) and 20p11.23 having the highest expression frequencies [216]. Moreover, the group discovered correlations between HERV-H expression and lymph node invasion as well as microsatellite instability (MSI) of tumors [216]. Interestingly, CRC-associated HERV-H sequences displayed activating histone marks in their 5′ LTR regions [216] and previous studies reported MYB (a proto-oncogenic transcriptional activator) binding to HERV-H LTRs, leading to a sevenfold increase in promoter activity [217]. In addition, the N-acetyltransferase 1 (NAT1) was shown to be crucial for self-renewal and neural differentiation of primed pluripotent stem cells, while loss of NAT1 resulted in a significant increase in the HERV-H transcript [218]. Furthermore, ERVH-2 performed well as a specific immunological target, since T cells stimulated with HERV-H-specific peptides resulted in proliferation of mostly CD8^+^ T cells, leading to increased lysis of CRC cell lines [219]. Additional to HERV-H expression, ERVW-1^ε^ levels were proposed as putative prognostic markers because of a statistically significant association with decreased overall survival in rectal cancer but not in patients with colonic cancer [220].

A link between chronic inflammation and cancer has been established for various tumors, and, specifically for CRCs, an inflammatory microenvironment has been recognized to be a cause, hallmark, and consequence of disease [221]. A subset of patients with colon cancer was shown to express a *HERV-H_9q24.1::IL33* fusion transcript required for tumor growth [222]. IL33 is a proinflammatory cytokine produced by epithelial and endothelial cells [223] that has been demonstrated to correlate in its expression with CRC progression and metastasis [224]. The particular function of the *HERV-H_9q24.1::IL33* product is unknown but, based on the function of native IL33, might include roles as a modified cytokine or nuclear factor regulating gene transcription [223]. Further, HERV LTR promoted chimeric transcripts detected specifically in CRC tissues and cell lines including the ion transporter *SLCO1B3^λπ^*, which is frequently mutated in CRC [222].

In contrast to fusion transcripts that result in aberrant cellular genes, TIP60 (HGNC: KAT5) has been described as a regulator of the inflammatory effects of HERV expression inside the cell [225]. KAT5 is a tumor suppressor that is found to be repressed in early stages of CRCs and breast cancers [225]. A publication by Rajagopalan et al. (2018) indicated that KAT5 downregulation results in increased levels of HERV expression and associated inflammatory responses [225]. In normal cells, KAT5 induces H3K9 trimethylating enzymes SUV39H1 and SETDB1 in a BRD4-dependent manner, which leads to global inhibitory methylation of HERV loci [225]. In KAT5-repressed cancer cells, the study investigators detected induction of IRF7 mediated by the intracellular pathogen sensing STING (HGNC: TMEM173), resulting in an inflammatory response and further tumor growth [225].

While viral gene products are readily detected by innate immune receptors, most cellular lncRNAs escape the surveillance mechanisms and, in this fashion, are able to interfere with regulatory pathways. For instance, the lncRNA *EVADR* on chromosome 6q13 was observed to be induced by the ERV1 LTR MER48 specifically in colon, rectal, lung, pancreas, and stomach adenocarcinomas [226]. In a similar way, the ERV1 LTR MER61C on chromosome 514.1 drives transcription of the lncRNA *PURPL* (*LINC01021*), which is increased in CRC cell lines and tumors [227,228]. While higher *EVADR* expression was associated with slightly decreased patient survival rates [226], CRC tumors with higher levels of *PURPL* RNA resulted in improved survival rates, and induced expression in CRC cell lines lead to increased chemosensitivity according to Kaller et al. (2017) [227]. While Kaller et al. showed a positive correlation between *PURPL* RNA and TP53 target expression [227], just months before, Li et al. (2017) observed suppressive effects of *PURPL* RNA on TP53 through the binding of MYBBP1A (a TP53 stabilizer), leading to reduced proliferation and tumor growth in a mouse model [228]. Despite the differing findings concerning the function and effects of *PURPL* RNA on TP53 levels, both publications confirmed the induction of *PURPL* by TP53 [227,228]. At this point, it should be mentioned that Kaller et al. did not divide their CRC patient population according TP53 mutation status when comparing *PURPL*^high^ and *PURPL*^low^ cohorts. Accordingly, *PURPL* levels could very likely be a surrogate for TP53 status, being 50% mutated in the analyzed CMS subtype 4 group, and hence not in themselves impact survival rates. Nonetheless, *EVADR* and *PURPL* lncRNAs do present potential prognostic markers.

Additional to the potential chemo-sensitizing action of the *PURPL* lncRNA, RRx-001, a novel anticancer drug in phase III clinical trials [229], displayed the induction of HERV-Fc2 *env* (HGCN: ERVFC1-1^ε^) and HERV-L LTRs in colon cancer cells resulting in an antiviral response, which sensitizes the cells to immuno-, radio-, and chemotherapy [230]. CRISPR-Cas9-mediated knockout of HERV-K (HML-2) env in DLD-1 colorectal cancer cells reduced migration, invasion, and tumor colonization and was sociated with nuclear protein-1 (NUPR1) reduction, indicating a potential connection [231]. Conversely, the expression of immunogenic HERVs was associated with immune checkpoint activation and microsatellite instability, a predisposition to mutation resulting from impaired DNA mismatch repair, in patients with colorectal adenocarcinoma, turning the induction of such HERVs into a potential target for therapy [71]. Lastly, tumors with high HERV-H level were shown to exhibit low levels of FOXP3^+^ T regulatory cells, rendering them more responsive to cytolytic targeting and treatment with immune checkpoint inhibitors [232].

Despite the spatial and physical difference between CRC and gastric cancer, several similarities can be found between the two. Analogous to *CREB5* in CRC, a HERV-H LTR on chromosome 17q21 was discovered to drive the expression of an alternative transcript of gasdermin-like (*GSDML*) in the stomach cancer cell line AZ521, while transcription through the cellular promoter was only detected in stomach tissues of healthy individuals [206]. Overexpression of *GSDML* has been reported to be associated with tumor progression and carcinogenesis [233]. On the contrary, the *TPTE* pseudogene on chromosome 22, containing a HERV-H element (*TPTEP1*), also termed *psiTPTE22-HERV*, was reported to be downregulated in gastric cancer compared to adjacent normal tissue samples [234]. The protein product of *TPTEP1* has been postulated to carry tumor suppressor functions, but these have not yet been confirmed [234].

## 10. HERVs in Liver Cancer—The Opening Chapter

Liver cancers are the third leading cause of cancer-related death in males worldwide [235]; however, only limited reports are available on the influence of HERVs in liver cancer oncogenesis. Ahn and Kim (2009) together with Liang et al. (2009) reported increased expression of HERV-H, HERV-R.3-1, and HERV-P in overall liver cancers without taking into consideration distinct cancer subtypes [236,237], while several other groups reported the distinct activation of HERV-K (HML-2) and HERV-P in hepatocellular carcinomas (HCCs) specifically [238,239]. However, a study by Liu et al. (2021) demonstrated that the LRP1B mutation was associated with the overexpression of HERV-H LTR-Associating 2 (HHLA2) in patients with HCC [231]. HCC has a high morbidity and constitutes more than three-fourths of all cases with liver cancer [62]. Interestingly, a large number of HERV LTRs, including LTR1, LTR12C, and THE1B, were found upregulated in human HCC tumors (Figure 7) [240]. HCC tumors with high LTR activation were associated with high risk of relapse [240,241], a more aggressive phenotype [240,241], poor prognosis [241], and impaired cell differentiation in animal models [16,242]. Moreover, HERV-K (HML-2) gene products, while only moderately elevated in patients [243], were described to positively correlate in their expression with tumor cell dedifferentiation, mortality rates, TNM stage, and cirrhosis in HCCs [239]. Furthermore, in a subgroup of patients with HCC antibodies against HERV-K (HML-2) Gag were found indicating the immunogenicity of the viral gene product [238,243]. Additional to its potential as prognostic marker and therapeutic target, HERV-K (HML-2) expression might also provide resolution to a dispute concerning the origin of a liver cancer cell line. It is still debated whether HepG2 cells are derived from HCC or hepatoblastoma.

The absence of any HERV-K expression in HepG2 cells points to the latter [244], especially since other HCC cell lines such as Hep3B display a HERV-K (HML-2) profile similar to HCCs [130]. Furthermore, ERVW-1^ε^ transcription was shown to be enhanced in HCC patients and cell lines, increasing cell proliferation, metastasis, and tumorigenicity [245]. ERVW-1^ε^ expression in HepG2 cells [246] was demonstrated to be further induced by Hepatis B virus (HBV), the most common cause of HCC, through HBx protein-mediated NF-κB activation [244]. In HCC cells, ERVW-1^ε^ actively induced the MEK/ERK pathway and suppressed doxorubicin-induced apoptosis via MEK/ERK cascade, highlighting its central role in drug resistance [245]. ERVW-1^ε^ also induces cell fusion [19,247] and carries an immunosuppressive function [248], both of which have been shown to perpetuate oncogenesis.

## 11. HERVs in Nervous System Cancers—The Wicked Side of HERV-W

Perhaps the best described and most widely accepted involvement of HERVs in pathogenesis has been in neurological diseases. Links between HERVs and MS, amyotrophic lateral sclerosis, and schizophrenia are supported by numerous publications and have been summarized well by Gruchot et al. (2019) [249] and Dolei et al. (2019) [250]. Interestingly, one specific HERV-K (HML-2)-derived SINE fragment was identified by random amplified polymorphic DNA to be absent in a patient with a grade IV glioblastoma (GBM) [251]. Even though this finding was unique to the patient and could not be observed in 32 other gliomas, the method might be useful to identify the absence of HERV-K (HML-2) sequences normally located in tumor suppressor genes, such as *BRCA2*, *XRCC1*, and *NBPFs*, as a marker for oncogenesis (Figure 8) [251,252]. On the contrary, a much more pronounced role in inflammatory neurological disease as well as brain cancers has been confirmed for HERV-W. Substances such as caffeine and aspirin that are able to pass the blood–brain barrier have been reported to increase ERVW-1^ε^ protein and HERV-W *Gag* mRNA as well as protein levels in human SH-SY5Y neuroblastoma cells [253]. The concentration for both substances correlated positively with HERV-W gene expression and negatively with cell survival [253]. Intriguingly, caffeine acted in a luciferase assay on the HERV-W promoter, while aspirin did not change the promoter activity [253]. Additionally, human cytomegalovirus—a betaherpesvirus implied to be present with high prevalence in many brain cancers [254,255,256,257,258]—was shown to induce upregulation of HERV-W alongside HERV-T, HERV-F, ERV-9, HERV-K (HML-2 to 4 and HML-7 to 8), and HERV-L elements in GliNS1 cells [259]. Although less pronounced than in the active infection, UV-inactivated virus was also able to stimulate HERV expression [259]. While it is assumed that the majority of HERV-W gene products are made by neurons and myeloid cells, i.e., monocytes and microglia, low expression of *ERVW-1^ε^* mRNA has been detected in normal astrocytes, which is increased up to twofold in U-87 MG astrocytoma cells (26–50% and 59–74%, respectively, compared to levels in placental tissues) [260]. Conversely, neuroblastoma cell lines SK-N-DZ and SK-N-AS have been described as reliable producers of HERV-W RNA [261,262]. This expression was shown to be further induced after recovery from hypoxia [261] or exposure to the antipsychotic drug valproic acid (VPA) [262]. VPA, a commonly prescribed medication for seizures and bipolar disorder, is a histone deacetylase inhibitor suggested to activate additional to HERV-W ERV9 promoters through chromatin remodeling in the neuroblastoma cell lines [262].

Through studies of neuroblastoma cell lines SH-SY5Y and IMR-32, ERVW-1^ε^ was demonstrated to stimulate the expression of *SK3* [263] and *TRPC3* [264]. Additionally, ERVW-1^ε^-mediated SK3 channel activation was identified to be dependent on the CREB1 and documented to result in an increased potassium ion (K^+^) current [263]. On the other hand, the TRPC3 channel was postulated to be activated through either the derepression of the TRPC3 inhibitor, *DISC1*, or the direct induction of *TRPC3* expression by ERVW-1^ε^ [264]. In addition to the Ca^2+^ and Na^+^ influx caused by the TRPC3 channel, studies in recent-onset schizophrenia demonstrated the capability of ERVW-1^ε^ to upregulate the Ca^2+^ induced K^+^ channel SK2, further escalating the potassium ion current [265]. While both findings point towards new functions of ERVW-1^ε^ especially in neuronal cells, experiments showing whether the fusogenic potential of ERVW-1^ε^ was confounding the ion influx and cell behavior observed were missing [263,264]. Accordingly, roles of ERVW-1^ε^ in the aforementioned functions have to be further evaluated.

Besides expression in neuroblastoma cell lines, HERV-W RNA alongside HERV-K (HML-2), -H, and -C RNA was found to be highly enriched in microvesicles of cells derived from glioblastoma tumor specimens [266]. These microvesicles were reported to carry the ability of horizontal gene transfer between cancer cells, potentially leading to increased levels of HERV proteins in neighboring cells [266]. HERV-K (HML-2) Env expression has been reported to be upregulated in human Merlin-negative schwannoma and in all meningioma grades via the CRL4 (DCAF1) and YAP/TEAD pathway [267]. In addition, C-MYC was shown to bind to the HERV-K (HML-2) LTR and lead to *env* expression, which in turn can be inhibited by SMARCB1 binding free C-MYC [268]. Biallelic SMARCB1 is characteristic of atypical Teratoid Rhabdoid Tumor (AT/RT), a rare pediatric central nervous system cancer [268]. HERV-K (HML-2) Env in turn increases JUN and pERK4/2 when overexpressed in Schwann cells [267], while retroviral protease inhibitors ritonavir, atazanavir, and lopinavir reduced proliferation of schwannoma and grade I meningioma cells [267]. Furthermore, ERVW-1^ε^ and ERVFRD-1^ε^ were demonstrated to bind to mitochondria of chemo-resistant U87RETO glioblastoma cells. allowing for their direct transfer across cell membranes [269]. This process was confirmed to utilize the fusion capabilities of the ERVW-1^ε^ and ERVFRD-1^ε^ and might have a major role in the resistance to mitochondria-targeted drugs [270]. Antibodies targeting ERVW-1^ε^ and ERVFRD-1^ε^ proteins were observed to inhibit the direct cellular uptake of mitochondria, signifying new strategies to combat drug resistance in glioblastomas [270].

## 12. HERVs in Prostate Cancer—The Dancing Partner of the Androgen Receptor

Prostate cancer is the most frequent cancer in males in Western countries and the second most common cancer in males worldwide [62]. Despite increased survival rates, curative treatments, such as surgery and radiation, convey serious side effects [62], so that active surveillance becomes the preferred approach for men with less-aggressive prostate cancer. Directed therapies targeting HERVs that are reportedly dysregulated in prostate cancer cell lines and tissues present a milder approach for prostate cancer treatment. A prime candidate in prostate cancer is HERV-K, which exhibits tight interactions with the testosterone receptor. While certain HERV-Ks, such as HERV-K17 on chromosome 17p13.1 (not HGCN: ERVK-17), show tissue-specific upregulation in prostate cells and downregulation in malignant cells [271,272], other members of the family, such as HERV-K-Mel (HML-6) on chromosome 16p11.1 [273] and HERV-K (HML-2) on chromosome 22q11.23 [238,274], display significantly higher expression in prostate cancer tissues and cell lines compared to healthy controls (Figure 9). This increased expression was identified to be androgen-dependent with several HERV-K (HML-2) LTRs containing predicted steroid hormone receptor-binding sites [274]. HERV-L LTR40a was shown to exhibit ligand-dependent recruitment of the androgen receptor (AR) functioning as an enhancer for *KLK3* [275]. KLK3 is the most well-studied biomarker for prostate cancer and is a model system to study androgen signaling [276]. As prostate cancer cell survival is highly dependent upon AR signaling, LTR40a induction might provide an alternative marker for disease detection. Additionally, inhibition of the LTR40a and its associated gene products could prevent the synthesis of HERV-regulated oncogenes. Reciprocal to the activation of HERV LTRs by the AR, HERV-K (HML-2) Rec has been documented to associate with the AR co-repressor, TZFP (HGNC: ZBTB32), relieving the TZFP-mediated repression of AR-induced transactivation [277]. Furthermore, HERV-K (HML-2) Rec binding of TZFP was reported to mitigate the direct transcriptional repression of the *MYC* gene promoter [277]. HERV-K (HML-2) Rec was also observed to inhibit the cytoplasmic negative regulator of the androgen receptor SGTA and lead to its translocation to the nucleus [278]. This in turn was shown to result in a higher sensitivity of the AR to its ligand in the cytoplasm and increased expression of AR-induced oncogenes as well as HERV-K (HML-2) gene products [278].

Targeted treatments might be especially beneficial for patients with advanced cancers, as HERV-K (HML-2) Gag antibodies are predictors of poor prognosis and correlate with prostate cancer progression [238,243,279]. The increased antibody titers might very well indicate higher HERV-K (HML-2) protein levels, but there are no publications that have analyzed both seropositivity and HERV protein levels in the same patients. Interestingly, increased HERV-K (HML-2) expression has not only been associated with higher levels of the HERV antigen, but also with recurrent gene fusions [272]. While over half of prostate cancers display fusions involving the oncogenic *ETS* transcription factors (including *ETS1*), *ETS1* has been described in several patients with prostate cancer involved in fusion with HERV-K17 [272] and HERV-K_22q11.3 [271,280].

Similar to the HERV-K (HML-2) Env-mediated regulation of cellular pathways shown in breast cancer cell lines, Ibba et al. (2018) were able to demonstrate that CRISPR-Cas9-facilitated knockout of HERV-K (HML-2) *env* leads to significant decrease in genes implicated in prostate cell transformation such as EGFR, NF-κB, SRSF1/2, and TDP-43 (HGNC: TARDBP) [281]. Furthermore, several HERV LTR-driven lncRNAs are reported to hijack key TFs with tumor suppressor functions, such as TP53 and SP1 [196]. In particular, the HERV9 LTR-driven lncRNA *SCHLAP1* was found to be upregulated in one-fourth of prostate cancers to act as an independent predictor of poor clinical outcomes and to play an essential role in tissue invasion and metastasis [49,282]. HERV-H provides another group of LTRs that function as promoter enhancers and lead to the transcription of lncRNAs [242]. Both activities are essential for human stem cells [242,283]—employing at times up to 40% of HERV-H sequences [17], although an aberrant increase has been associated with cancer stem cell formation in many malignancies, one being prostate cancer [284,285]. Additionally, single-cell transcriptome analysis revealed LTR7/HERV-H network activation to be significantly increased in localized and metastatic prostate cancers compared to normal prostate, making it a putative treatment target [286].

## 13. HERVs in Lung Cancer—The Love for Long Noncoding RNAs and Pseudogenes

Even though lung cancers are by far the deadliest cancers worldwide [62] and carry an equally high mutational burden as melanomas [122], only limited research on the relationship between HERVs and pulmonary diseases has been published. Being one of the leading causes of mutations in lung cancer, smoking was shown to induce HERV-K (HML-2 and 6) *pol* expression in multiple tissues (Figure 10) [287,288]. Insertional polymorphisms (absence and presence of a HERV sequences at a specific locus) can alternatively lead to differences in numerous regulatory elements at a specific locus and therefore have gained increased interest in the research community. Kayho et al. (2013) were able to demonstrate that HERV-K (HML-2)_soloLTR(1p13.2) homozygosity in never-smoker women is statistically associated with increased susceptibility to lung adenocarcinoma [288]. This finding is supported by the observation of a peptide-carrying sequence homology with HERV-K10 LTR specifically expressed in human lung adenocarcinoma A549 and absent in non-transformed fibroblasts [289]. The Krüppel-associated box domain-containing zinc-finger family protein (KZFP) is a transcriptional suppressor, which when expressed in cancer cells alters the expression of genes expression of genes related to the cell cycle and cell-matrix adhesion and suppresses cellular growth, migration, and invasion abilities [290].

Pan-cancer analysis revealed an association between HERV and KZFP expression [290]. In particular, CRISCR-Cas9-mediated knockout of the HERV-enhancer 1 on chromosome 12q24.33 (MER21B) and HERV-enhancer 2 on chromosome 19q13.43 (HERVK3-int, HGCN: ERVK3-1) resulted in decreased KZFP expression in lung adenocarcinoma (LUAD) cancer cell line A549 [290]. Overall increased KZFP expression was associated with better prognosis and lower cancer stage in patients with LUAD [290], while another study demonstrated less favorable outcomes linked to increased HERV—in particular HERV-E, HERV-L, HERV-H, and ERV3 expression [291]. In general, HERV-E transcripts expressed in lung cancer were among the strongest predictors of outcome, whereas MER21 was associated with ANKLE2 expression, LTR3 from HERV-K3I (HML-6) with ZNF8 expression, and THE1D-int with ZNF75D expression [291]. Another pan-cancer study revealed several single nucleotide variants affecting KIR2DL1, KIR2DL1 (MST/MaLR) downregulation associated with lung squamous cell carcinoma (LUSC) [292].

Intriguingly, in recent years, lncRNAs increasingly appeared to occupy a central position in the regulation of lung cancers. As first described, the ERV1 LTR MER48-driven lncRNA *EVADR* was detected in over 20% of patients with lung adenocarcinoma but undetected in normal lung tissues [226]. A combined statistical analysis of lung, pancreas, colon, rectal, and stomach adenocarcinomas indicated a significant association of *EVADR* expression with decreased survival rates [226]. Similarly, the regulatory lncRNA *HCP5* was found overexpressed in lymph node metastasis of small cell lung cancer [293,294]. The *HCP5* antisense HERV16 transcript in turn has been shown to interact with immune and regulatory checkpoints in various other cancers [295,296]. More interestingly, the *TPTE* pseudogene, containing a HERV-H element (*TPTEP1*), is downregulated in lung cancer compared to adjacent normal tissue samples [237]. It is known that in healthy tissues, the pseudogene is translated with the majority of the HERV sequence being spliced out, but the function of the resulting protein is still unknown [237]. For other roles of HERVs in lncRNA regulation, see HERVs in Other Genital Cancers (Ovary Cancer, Choriocarcinoma, and Endometrial Cancer)—The Ascent of New Possibilities. Supporting the potential use of ERVW-1^ε^ in tumor therapies (see HERVs in Leukemia—The Lifesavers for Cancer Cells), Lin et al. (2010) were able to induce syncytia formation and, in this way, reduce cell viability and tumor growth by transfecting *ERVW-1*^ε^ expression plasmids in human non-small cell lung cancer cells in vitro or by directly injecting the protein into tumors in mice [219,297]. Lee et al. (2016) reported that RNAi-mediated knockdown of *ERV3-1* in radioresistant A549 cells increased radiosensitivity and induced apoptosis, suggesting its potential use as a drug target for new anticancer therapeutics [298].

## 14. HERVs in Cancers of the Urinary System (Kidney and Bladder Cancer)—The Future Fire Fighters

While most HERVs operate below immune detection, research has shown that upregulation of HERVs in transformed cells can serve as a physiological tumor recognition signal, preventing the propagation of cancerous cells in early stages [117,140,203,230]. In advanced-stage cancers, such tumor suppressive functions are disrupted on multiple levels, one being through immune checkpoint activation [299]. Hence, newly developed immune checkpoint inhibitors have proven to be effective regimens for persistent cancers, especially for clear cell renal cell carcinoma (ccRCC), where clinically significant and robust responses have been observed [300]. To further enhance antitumor immune responses upon immune checkpoint blockade, Panda et al. (2018) examined HERVs as prospective inducible targets in patients with ccRCC [71]. The study investigators determined the subset of potentially immunogenic HERVs (piHERVs) with the greatest potential to induce immune responses, such as immune infiltration, higher cytotoxic T-cell levels, and M1 macrophage abundance (Figure 11) [71]. Despite lower overall survival of patients with higher expression of such HERVs, piHERV^high^ patients treated with immune checkpoint inhibitors experienced significantly improved prognosis and treatment responsiveness compared to their piHERV^low^ counterparts [71]. Out of all piHERVs, HERV-R.3-2 *env* (*ERV3-2*) expression was particularly increased in responders compared to non-responders, highlighting its tumor suppressive functions mentioned for other cancers (see HERVs in Lymphoma—The Silent Inducers and HERVs in Other Genital Cancers (Ovary Cancer, Choriocarcinoma, and Endometrial Cancer)—The Ascent of New Possibilities) [71]. Interestingly, similar results have been observed for patients with urothelial cancer who displayed high piHERV expression [301]. Solovyov et al. (2018) confirmed the positive correlation of piHERV levels with overall survival, progression-free survival, and response to immune checkpoint inhibitors for urothelial cancer [301]. In this case, *ERV3-1* was the most highly expressed HERV in responders [301].

Several underlying mechanisms have been proposed to drive the inherently higher expression of piHERVs. On the genetic level, piHERV^high^ patients demonstrated an enrichment of mutations in the *BAP1* gene, which is a deubiquitinase known to functionally associate with chromatin modulators [71,302]. The functional disruption of BAP1 may thus lead to chromatin remodeling resulting in piHERV expression [71]. A subgroup of patients with ccRCC was found to express HERV-E in high amounts driven by a hypomethylated LTR [303]. Inactivation of the von Hippel–Lindau tumor suppressor gene (*VHL*) with subsequent stabilization of hypoxia-inducible transcription factors, HIF1A and -2A, was shown to induce expression of two HERV-E transcripts on chromosome 6q15 (*HERV-E.CT-RCC-8* (HGCN: ERVE-4) and *HERV-E.CT-RCC-9*) through the HIF response element located in the viral LTR [303]. In other tumors and matched normal tissues, a hypermethylated LTR was reported to prevent the induction of the HERV-E.CT-RCCs [303]. Interestingly, recognition the HERV-E.CT-RCC-1 antigen, which carries a shorter sequence of the common region of *HERV-E.CT-RCC-8* and *-9*, by T cells was associated with the regression of human kidney cancer following unrelated allogeneic stem cell transplantation, suggesting its potential as a target for antitumor therapies [88]. Strikingly, ERVE-4 expression was also able to predict the response to anti-PD-1 in metastatic ccRCC [304]. Additionally, the observation that ERVE-4- and hERV47000-derived epitopes were able elicit a tumor-restricted CD8^+^ T-cell response were the basis for an ongoing phase 1 trial evaluating the safety and efficacy of the infusion of ERVE-4 TCR-transduced CD8^+^/CD34^+^-enriched T cells (NCT03354390) [304]. In urothelial cell carcinoma (UCC), effects mediated by HERV-E have been found beneficial and detrimental dependent on the specific virus. For instance, HERV-Ec11 on chromosome 11p15.4 (HGCN: ERVE-2) was found to be expressed only in a subgroup of patients with UCC, while HERV-Ec8 on chromosome 8p23.1 (HGCN: ERVE-3) was detected only in non-malignant urothelial tissues [305]. Furthermore, HERV-Ec1 encoded on chromosome 1q31.1 has been observed to be located in antisense orientation to a cytosolic phospholipase A2 group IVA (*PLA2G4A*) that is dysregulated in many human tumors, and thus theorized to contribute to fine tuning of PLA2G4A and to have a potential role in oncogenesis of UCC [305] (see also HERVs in Testicular Cancer—The Governors of Tumor Suppressor Genes).

Tobacco smoking has been recognized as an environmental risk factor for many cancers and is considered to be a main cause of bladder cancer [287,306]. HERV-E, -K, and -T have been indicated to be increased in the urothelium and a cell culture model of current smokers compared to non-smokers, making tobacco use a potential external factor related to HERV induction [287]. Additionally, HERV-K (HML-2) LTRs were reported to be strongly methylated in normal tissues and significantly hypomethylated in urothelial carcinomas resulting in higher HERV-K (HML-2) mRNA levels [141,307]. Other genetic risk factors for UCC include two mutations in the HERV-W LTR (142T > C and 277A > G) that have been identified to result in a new MYB-binding site [308]. Expression analysis confirmed that increased MYB binding at these sites caused significantly higher expression of *ERVW-1*^ε^, leading to amplified proliferation and viability of immortalized human uroepithelial cells [308]. Furthermore, pan-cancer analysis revealed several single nucleotide polymorphisms within HERVs significantly affecting ZNF99 [292]. Kidney cancer patients with a C2270G mutation in *ZNF99* (HERV-W/HERV17/LTR17) were shown to have significantly lower survival [292]. In addition, *ERVW-1^ε^* overexpression was significantly associated with a late-stage and aggressive-form UCC in patients, suggesting that it may impact the degree of malignancy in UCC [308].

The lncRNA urothelial carcinoma associated 1 (UCA1) is one of the best studied purely retroviral HERV gene products with major effects on cancer progression, tumor growth, apoptosis, invasion, radioresistance, chemoresistance, and metabolism [309]. UCA1 is a splice product of three exons located on chromosome 19p13.12 [310] with three isoforms (1.7 kb: UCA1, 2.2 kb: UCA1a or CUDR, 2.7 kb: UCA1b) [311,312]. UCA1 was originally discovered to be upregulated in bladder transitional cell carcinoma (TCC) by Wang et al. in 2006 [310]. Meanwhile, it has been confirmed to be activated primarily during normal gestation with persistent expression in the heart and spleen [313] and to be reactivated in every cancer type reviewed in this very report with the exception of lymphomas [309,312]. As it would exceed the scope of this review, we recommend the comprehensive summaries on the functions of UCA1 as regulator of miRNAs by Xuan et al. (2019) [309], as chromatin remodeler in complex with other proteins by Neve et al. (2018) [314], as a mediator of chemoresistance by Wang et al. (2017) [315], and as a potential biomarker by Xue et al. (2016) [312].

## 15. HERVs in Endocrine Cancers (Pancreas and Thyroid Cancer)—The Unknown Potential

Despite their rarity, endocrine tumors, particularly thyroid cancers, have dramatically increased in their incidence worldwide in the last four decades [316,317]. Additionally, anaplastic thyroid cancer [318] and adrenocortical carcinoma [319] have incredibly short survival times due to their aggressive nature, while pancreatic ductal adenocarcinoma is often associated with increased risk for metastasis and high mortality rates due to a lack of symptoms in early stages [320,321]. In pancreatic cancer, HERV-K (HML-2) has been suggested as a major tumor driver. Li et al. (2017) reported HERV-K (HML-2) *env*, *gag*, and *np9* genes as well as HERV-K (HML-2) Env protein expression to be significantly increased in pancreatic cancer cell lines as well as patients with pancreatic cancer compared to healthy controls (Figure 7) [322]. The RNAi-mediated knockdown of HERV-K (HML-2) *env* reduced proliferation and colony formation of the cancer cell lines and resulted in decreased tumor growth and metastasis in an in vivo mouse model [322]. Pathway analysis by Li et al. revealed an activation of the RAS/MEK/ERK pathway and inhibition of TP53 by HERV-K (HML-2) in pancreatic cancer cells [322], which was confirmed in breast cancer cells by Lemaître et al. (2017) (see Results HERVs in Breast Cancer—The Rise of New Biomarkers) [77]. Furthermore, the cleavage of S100A4 by the HERV-K (HML-2) protease was associated with cell cycle progression, differentiation, and metastasis in pancreatic cancer [323]. Moreover, Li et al. detected increased expression of *ERV3-1* and *ERV3-2* transcripts in prostate cancer cell lines [322]. These transcripts displayed an intact open reading frame in most cell lines, potentially giving rise to active Env proteins [322]. Interestingly, Li et al. also observed the release of viral-like particles with increased RT activities by pancreatic cancer cell lines Panc-1 and Panc-2, suggesting the potential horizontal transfer of viral proteins [322].

In addition to HERV-K (HML-2) Env, HERV-H Env was described to contribute to the oncogenesis of pancreatic cancers. HERV-H on chromosome 3q26 is one of the few from the family that contain a complete open reading frame for the Env protein (HGCN: ERVH-9^ε^) [324]. This ERVH-9^ε^ protein, also called Env60, has been upregulated in pancreatic cancer cells undergoing epithelial-to-mesenchymal transition (EMT), while also expressed in healthy pancreatic tissue [325]. Particularly, the immunosuppressive portion of the ERVH-9^ε^ protein amplified EMT and induced *CCL19* expression, which significantly correlated with the recruitment of immunosuppressive cells in patients [325]. In contrast, RNAi-mediated knockdown of HERV-H significantly decreased tumor invasion, CCL19 production, and the recruitment of immunosuppressive cells without affecting cell proliferation, suggesting the central role of ERVH-9^ε^ in immune regulation [325]. In another study, HERV-H_Xp22.3 (HGCN: ERVH-2) was reported to be upregulated in 2 of 12 pancreatic cancers [326]. However, ERVH-2 showed higher correlations with colorectal cancers being overexpressed in almost half the tumors (see also HERVs in Other Genital Cancers (Ovary Cancer, Choriocarcinoma, and Endometrial Cancer)—The Ascent of New Possibilities) [216,237,326]. Nonetheless, functional consequences of the overexpression of ERVH-2 must still be determined.

Unique to endocrine tissues is the high baseline expression of HERVs in non-malignant cells. For instance, *ERVW-1^ε^* was demonstrated to exhibit constitutive expression in normal pancreatic tissues, while reduced expression was detected in pancreatic adenocarcinoma [327]. Furthermore, two specific HERV-E transcripts derived from chromosome 17q11 were solely detected in healthy pancreatic and thyroid tissues and absent in other healthy cells (Figure 7) [328]. Particularly, the thyroid gland was shown to display high HERV expression levels in normal thyroid tissue, indicating possible physiological functions [329]. Interestingly, the thyroid is the only human tissue to express *envT* (HGCN: *ERVS71-1^ε^*), an *env* gene not produced in the placenta [330]. As such, *ERVS71-1^ε^* is an exceptional candidate to analyze the potential oncogenic roles of SNPs in the development of thyroid cancers as it shows a lack of conservation in primates and no implied essential physiological functions [331]. Along these lines, an analysis of disease-associated SNPs by Wallace et al. (2018) discovered several polymorphic HERV-K (HML-2) insertion sites uniquely associated with thyroid malignancies [332]. Further supporting the potential oncogenic role of polymorphic HERVs is the finding of the RT inhibitor nevirapine decreasing the proliferation and increasing the differentiation of human thyroid anaplastic carcinoma reported in a case study [333,334,335]. In contrast to its beneficial effects in ccRCC (see HERVs in Cancers of the Urinary System (Kidney and Bladder Cancer)), HHLA2 expression was observed to be associated with poor survival in patients with papillary thyroid cancer (PTC) [336]. HHLA2 displayed a tumor promoter function that enhanced the proliferation of PTC cells [336]. Altogether, endocrine cancers display unique features that call for additional evaluation to further expand the understanding of the relationship between HERVs and cancer.

## 16. HERVs in Other Cancers (Osteosarcoma, Head and Neck Squamous Cell Carcinoma)—The Hodgepodge of Hope for Novel Therapies

Rare cancers, by definition, only provide limited case numbers for investigation. Accordingly, only single reports of HERVs evaluated in such cancers are available. Despite a high incidence in children, osteosarcoma is a rare malignancy in adults [337]. We identified only a single study on human osteosarcoma reporting the statistically significant upregulation of 35 and downregulation of 47 HERV mRNAs in osteosarcoma tissues compared to healthy controls [337]. The most significant HERV elements differentially expressed included LTRs of the HERV-L, HERV-K (HML-2), and ERV-1 [337]. The study lays the foundation for the identification of tumor-specific viral target for vaccine strategies. Proof of concept for such a strategy is the AH1 peptide (a murine endogenous retroviral (MuERV) envelope protein-derived peptide) that has been documented as a potent vaccine against murine WEHI-164 fibrosarcoma, eliciting a potent CD8^+^ T-cell response [338,339]. In addition, MuERV envelope glycoprotein gp90 has been shown to be tumor specific to murine colon carcinoma [340].

Head and neck squamous cell carcinomas (HNSCC) comprise 90% of all head and neck cancers and are relatively common [341,342]. HNSCCs are often inoperable due to the complex anatomy, making radio- and chemotherapy the only option [343]. Accordingly, radioresistance poses a major problem resulting in very low survival rates [62,343]. Findings by Michna et al. (2016) documenting an induction of *ERV3-1* and ERVMER34-1 *env* upon exposure of HNSCC cell lines to γ-radiation indicate a potential target to overcome radioresistance (Figure 10) [341]. The study investigators identified putative interactions of induced ERV3-1 with genes associated with radiation response, such as GPCR signaling, transmembrane transport of small molecules, generic transcription pathway, signaling by Rho GTPases, DNA repair, CD28-dependent Pi3K/Akt signaling, and cell cycle [341] This hypothesized role of ERV3-1 is further supported by a study reporting increased levels of ERV3-1 in radioresistant A549 lung cancer cells but not in less radioresistant H460 cells [344]. The same study observed an increase in radiosensitivity and apoptosis upon RNAi-mediated knockdown of *ERV3-1* in A549 cells [344].

## 17. Discussion of Novel Options for Cancer Treatment Facilitated by HERVs

As described in this review, many studies have demonstrated increased HERV levels in tumor cell lines and tumor tissues compared to normal healthy tissues, suggesting two potential treatment approaches. On the one hand, strategies have been proposed that target pathways in which HERVs are involved [234,345,346], as outlined for various cancers above. HERVs might provide another pharmacological target in this way. Moreover, HERV-derived HERV restriction factors such as suppressyn (HGCN: ERVH48-1), a HERV-F-derived inhibitor of ERVW-1^ε^-mediated fusion, might serve as a starting point for drug development. However, discoveries of HERV genes and LTRs involved in regulatory mechanisms are very new and still advancing with the recent development of more accurate and affordable sequencing techniques. Accordingly, we expect to see the discovery of more detailed cancer-specific signaling networks that include HERVs in the near future.

On the other hand, treatment strategies targeting HERV proteins as tumor-specific antigens have been suggested [83,84], assuming HERV expression is a consequence of transcriptional changes in tumors. HERVs as cancer-specific antigens in hematological cancers appear to be particularly promising. Saini et al. (2020) found HERV-specific T cells are present in 17 of the 34 patients with leukemia, recognizing 29 HERV-derived peptides representing 18 different HERV loci, among which ERVH-5, ERVW-1, and ERVE-3 had the strongest responses [347]. Furthermore, the ancestral retroviral HEMO envelope gene (Human Endogenous MER34 ORF) is hailed as a pan-cancer target for leukemia, lung, adrenal, thyroid, breast, ovarian, uterus, cervical, prostate, esophagus, stomach, colon, liver, pancreas, renal, bladder, brain, and skin cancer [348]. Vaccinations of mice with HERV epitopes were shown to be safe and able to generate tumor-specific immune cells [219,339,349,350]; however, the cancer risk-reducing properties of human vaccines contaminated with HERVs have yet to be confirmed or associated with HERVs [351]. Nonetheless, the advent of CAR-T cells [169] and checkpoint blockade inhibitors [139] might offer additional options to enhance tumor-specific immunogenicity. The generation of HERV-specific T cells recognizing tumors was demonstrated in a proof-of-concept study by Bonaventura et al. (2022), gleaning further enthusiasm for the potential use in targeted therapies [352].

Furthermore, HERV-H LTR-associating proteins 1 and 2 (HHLA1 and HHLA2) on chromosomes 3q13.13 and 8q24.22 have been shown to carry immune checkpoint functions [353]. First described by Mager et al. in 1999, *HHLA1* and *HHLA2* are both members of the B7 family and obtain their polyadenylation signal through HERV-H LTR regions [354,355], thus revealing a control mechanism of viral origin [356]. While both proteins are part of oncogenic signaling pathways, HHLA2 has been detected in several human cancers [195]. HHLA2 was found to be overexpressed in basal breast cancer [357], triple-negative breast cancer [357], colorectal cancer [358], lung cancer [359,360,361], liver cancer [362], bladder urothelial carcinoma [363], ccRCC [364,365,366], pancreatic cancer [367,368,369], osteosarcoma [370], oral squamous cell carcinoma [371], and many other cancers [357,364] compared to adjacent normal tissue or healthy controls. Additionally, elevated HHLA2 protein levels were associated with tumor size, tumor stage, lymph node metastasis, and low relapse-free and overall survival in these cancers [357,358,359,360,361,362,363,364,365,366,367,368,369,370,371]. Thus, HHLA2 might not only become a suitable prognostic marker, as shown by Zhang et al. (2021) in a Chinese cohort [372], but also a potential immunotherapy target for patients who do not respond to other immune checkpoint inhibitors, such as PD-1 inhibitors. Surprisingly though, HHLA2 mRNA levels in the blood were described by Shimonosono et al. (2018) to be downregulated in patients with gastric cancer [373]. Furthermore, lower *HHLA2* expression in the blood of patients with gastric cancer had a significant correlation with the depth of tumor invasion, poorer survival rates, distant metastasis, and tumor-node-metastasis (TNM) stage [373]. Correspondingly, HHLA2 expression in renal cancers was identified as higher in low pathological grades than in high pathological grades, suggesting its protective capabilities [364]. In addition, hypomethylation of HHLA2 was indicative of a more favorable outcome [364]. First attempts to use HHLA2 as treatment target demonstrated siRNA-mediated knockdown of HHLA2, resulting in the inhibition of NSCLC and reduced HCC in mice [364]. Additionally, Bhatt et al. (2020) were successful in developing HHLA2-targeting antibodies that specifically block its immunoinhibitory activity [374]. Overall, HHLA2 is a promising target with wide applicability.

In 1997, Perron et al. first described retroviral particles, which were later termed MS-associated retroviruses (MSRVs), in the leptomeningeal tissue of MS patients [375]. Subsequent studies of the MSRV confirmed high homology to the HERV-W family, which could be explained by the recombination of a HERV-W sequence on chromosome Xq22.3 and another defective HERV-W *env* on chromosome 5 [376]. Functional studies of the MRSV-Env protein, frequently detected in the blood and lesions of MS patients, identified it as having strong agonist functions on Toll-like receptor 4 (TLR4), leading to neuroinflammatory effects [377]. Accordingly, the Institut Mérieux group, INSERM, and GeNeuro developed the MSRV-Env-targeting antibody GNbAC1 (now, Temelimab), which recently completed phase IIb clinical trials for the treatment of MS [1]. Clinical trials showed minimal side effects and good tolerance [378] as well as clinical improvements in patients with MS [1]. While Temelimab failed to show an effect on features of acute inflammation, it demonstrated preliminary radiological signs of possible anti-neurodegenerative effects for patients who had taken the highest dose [1]. Patients had less T1-hypointense lesion (associated with MS disability and progression), a reduction in brain tissue loss, and improvement in MRI markers of remyelination, suggesting that Temelimab might promote remyelination and prevent loss of nerves [1]. Additionally, Temelimab proved to be well-tolerated in a phase I trial for the treatment of type I diabetes [5]. As mentioned above, ERVW-1^ε^, also known as Syncytin-1, is currently the most promising HERV-related cancer therapeutic target. While clinical trials of Temelimab modeled the capabilities of HERV-W-targeting drugs in general, cross-reactivity studies of Temelimab in particular indicated binding of the antibody to ERVW-1^ε^ at high concentrations, suggesting the possible use of Temelimab in cancers [379]. In conclusion, clinical studies of Temelimab show promising results laying the foundation for development and use of HERV-targeting antibodies in other diseases such as cancers.

In summary, HERVs are involved in various homeostatic and pathogenic pathways with potential effects on cancer development and progression. The usage of HERVs themselves as therapeutic agents, as well as the HERV proteins as tumor-specific targets, are promising but must be further evaluated to exclude any undesired side effects. Therefore, this review aimed to comprehensively overview HERV involvement in different cancers to provide a summary of the research possibilities.

## Figures and Tables

**Figure 1 biomedicines-11-00936-f001:**
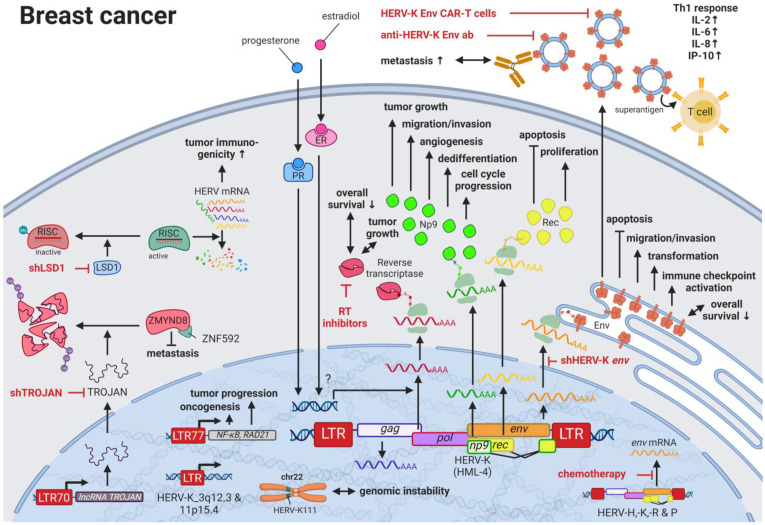
The role of HERVs in breast cancer. Evaluated treatments are marked in red. Abbreviations: Th1 = T helper cell 1, ER = estradiol receptor, PR = progesterone receptor, LTR = long terminal repeat, gag = group antigen (capsid), pol = polymerase, RT = reverse transcriptase, env = envelope. If not otherwise stated HERV-K = HML-2.

**Figure 2 biomedicines-11-00936-f002:**
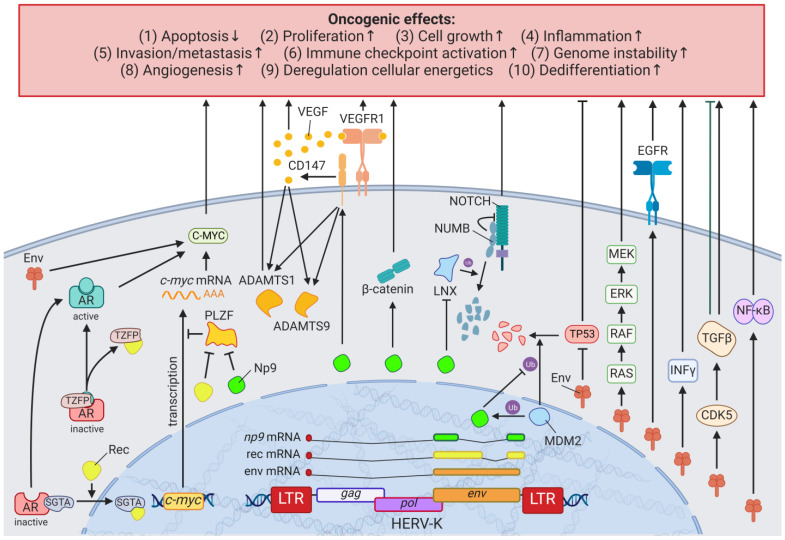
The effects of HERV-K (HML-2) Np9, Rec, and Env proteins on oncogenesis. LTR = long terminal repeat, gag = group antigen (capsid), pol = polymerase, env = envelope, HERV-K = HML-2.

**Figure 3 biomedicines-11-00936-f003:**
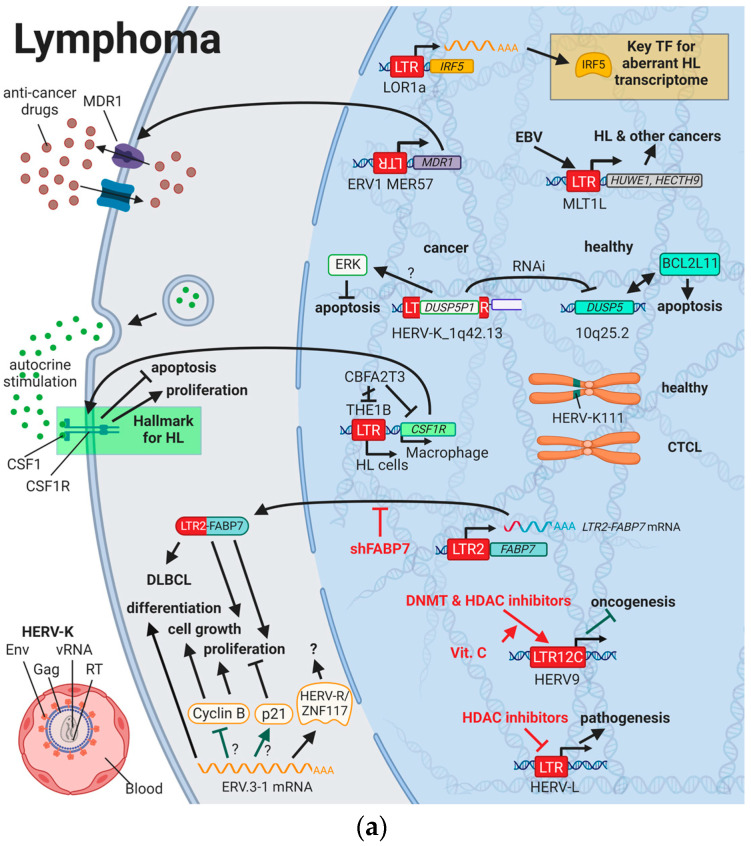
The role of HERVs in (**a**) lymphoma and (**b**) leukemia. Evaluated treatments are marked in red. Abbreviations: TF = transcription factor, HL = Hodgkin’s lymphoma, EBV = Epstein–Barr virus, CTCL = cutaneous T-cell lymphoma, DLBCL = diffuse large B-cell lymphoma, DNMT = DNA methyltransferases, HDAC = histone deacetylases, LTR = long terminal repeat, CML = chronic myelogenous leukemia, AML = acute myelogenous leukemia, B-CLL = B-cell chronic lymphocytic leukemia, vRNA = viral RNA, RT = reverse transcriptase, Env = envelope protein, TM = transmembrane domain, shNp9 = siRNA targeting np9, shFABP7 = siRNA targeting FABP7. If not otherwise stated HERV-K = HML-2.

**Figure 4 biomedicines-11-00936-f004:**
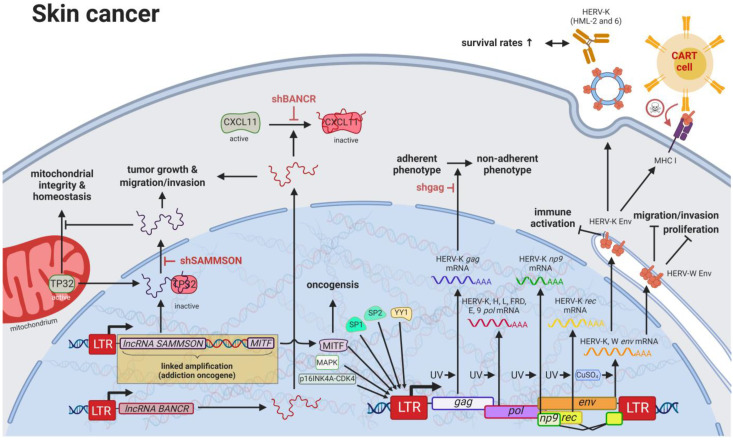
The role of HERVs in skin cancer. Evaluated treatments are marked in red. Abbreviations: LTR = long terminal repeat, gag = group antigen (capsid), pol = polymerase, RT = reverse transcriptase, env = envelope, lncRNA = long non-coding RNA, shBANCR = siRNA targeting BANCR. If not otherwise stated HERV-K = HML-2.

**Figure 5 biomedicines-11-00936-f005:**
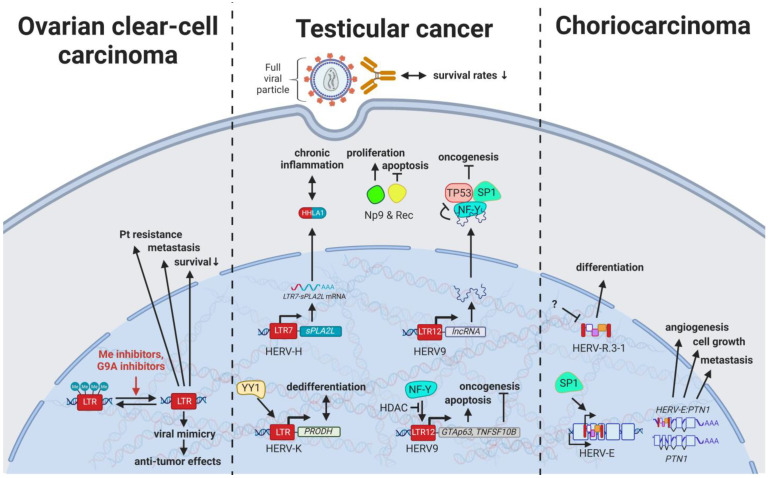
The role of HERVs in genital cancers. Evaluated treatments are marked in red. Abbreviations: HDAC = histone deacetylases, Me = methylases, G9A = G9a methyltransferase, Pt = platinum treatment, HERV-K = HML-2.

**Figure 6 biomedicines-11-00936-f006:**
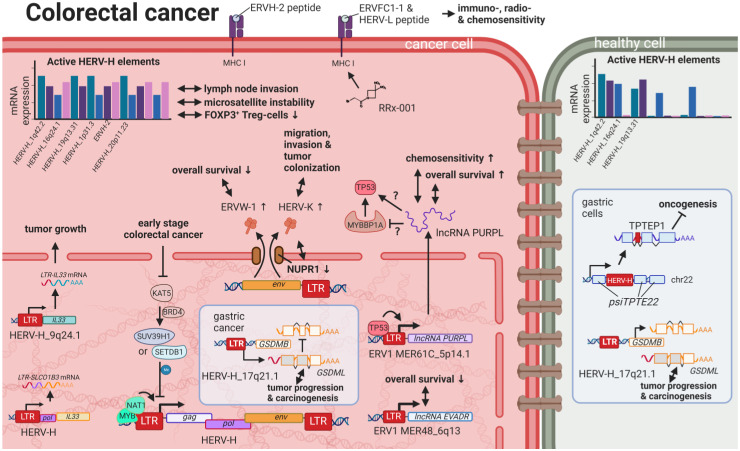
The role of HERVs in colorectal cancer. Evaluated treatments are marked in red. Abbreviations: lncRNA = long non-coding RNA, LTR = long terminal repeat, gag = group antigen (capsid), pol = polymerase, env = envelope, HERV-K = HML-2.

**Figure 7 biomedicines-11-00936-f007:**
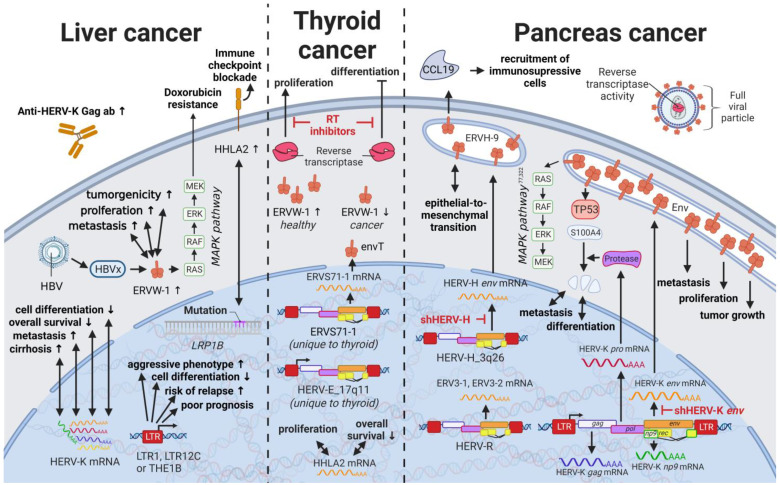
The role of HERVs in liver and endocrine cancers. Evaluated treatments are marked in red. Abbreviations: HBV = hepatitis B virus, MAPK = MAP kinase, ab = antibody, LTR = long terminal repeat, gag = group antigen (capsid), pol = polymerase, RT = reverse transcriptase, env = envelope, HERV-K = HML-2.

**Figure 8 biomedicines-11-00936-f008:**
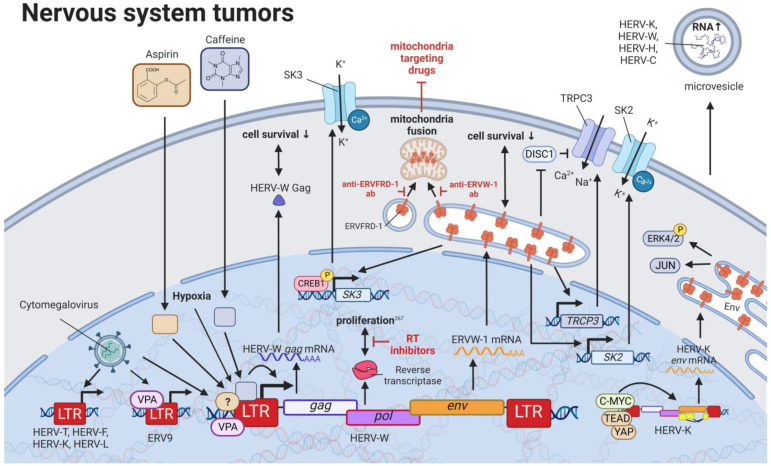
The role of HERVs in nervous system tumors. Evaluated treatments are marked in red. Abbreviations: K^+^ = potassium, Ca^2+^ = calcium, ab = antibody, LTR = long terminal repeat, gag = group antigen (capsid), pol = polymerase, RT = reverse transcriptase, env = envelope, HERV-K = HML-2.

**Figure 9 biomedicines-11-00936-f009:**
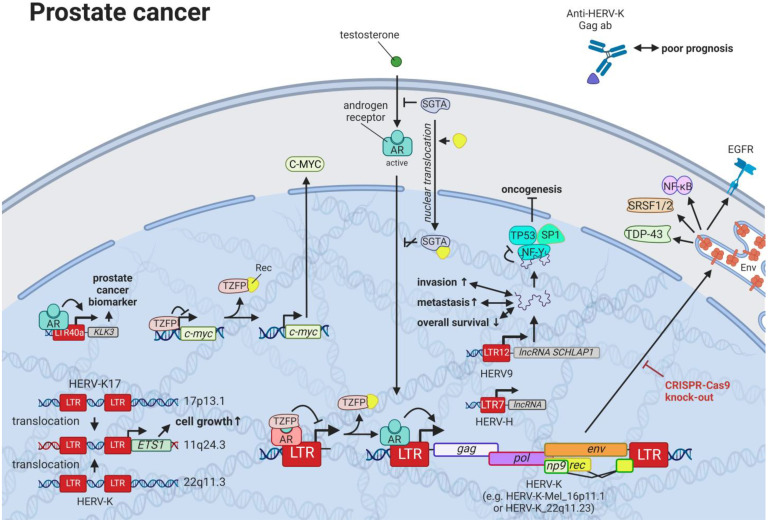
The role of HERVs in prostate cancer. Evaluated treatments are marked in red. Abbreviations: AR = androgen receptor, ab = antibody, lncRNA = long non-coding RNA, LTR = long terminal repeat, gag = group antigen (capsid), pol = polymerase, RT = reverse transcriptase, env = envelope, HERV-K = HML-2.

**Figure 10 biomedicines-11-00936-f010:**
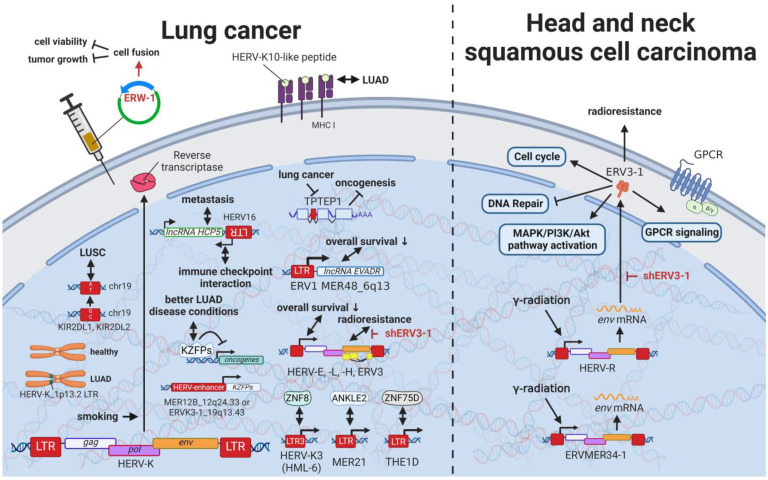
The role of HERVs in lung, head, and neck cancer. Evaluated treatments are marked in red. Abbreviations: LUAD = lung adenocarcinoma, LUSC = lung squamous cell carcinoma, GPCR = G-protein-coupled receptor, LTR = long terminal repeat, gag = group antigen (capsid), pol = polymerase, env = envelope. If not otherwise stated HERV-K = HML-2.

**Figure 11 biomedicines-11-00936-f011:**
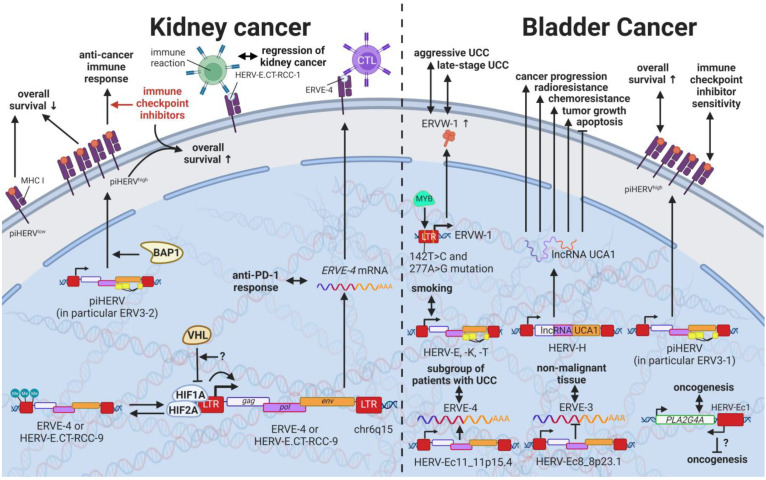
The role of HERVs in cancers of the urinary system. Evaluated treatments are marked in red. Abbreviations: CTL = cytotoxic T cell, UCC = urothelial carcinoma, RCC = renal cell carcinoma, piHERV = potentially immunogenic HERVs, LTR = long terminal repeat, gag = group antigen (capsid), pol = polymerase, env = envelope, HERV-K = HML-2.

**Table 1 biomedicines-11-00936-t001:** Annotations used in this review. Greek letters serve as abbreviation and denote complete or partial viral origin of a gene, RNA, or protein.

Annotation	Meaning
*gene^λ^*	gene consists of a viral LTR (λ)
*gene^γ^*	gene consists of a viral *gag* (γ) gene
*gene^π^*	gene consists of a viral *pol* (π) gene
*gene^ε^*	gene consists of a viral *env* (ε) gene
*HERV:gene* ^λγπελθ^	gene is a viral (λγπελ)–human (θ) gene fusion (see above)
*HERV::gene*	alternative isoform of a gene induced by a viral LTR

## Data Availability

Not applicable.

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
