# Peer review of "HERVs and Cancer—A Comprehensive Review of the Relationship of Human Endogenous Retroviruses and Human Cancers"

_biomedicines, 2023, doi:10.3390/biomedicines11030936_

Round 1
Reviewer 1 Report
Human endogenous retroviruses (HERVs), once external pathogens, have substantial functions. Recent studies show that HERVs are closely related to some diseases, including cancers, schizophrenia, and autoimmune diseases. In this review, the author summarized the effects of HERVs on 15 different cancer categories including solid tumors, lymphomas, and leukemias. This is an interesting study. But it needs some minor revision before published.
1. In the “Introduction”, the authors should introduce the relationships between HERVs and some other diseases. See the reference:
Multiple sclerosis: Mult Scler. 2022 Mar;28(3):429. doi: 10.1177/13524585211024997; Mult Scler. 2015 Jun;21(7):885. doi: 10.1177/1352458514554052; Mult Scler Relat Disord. 2022 Jan;57:103383. doi: 10.1016/j.msard.2021.103383.
Schizophrenia: Int J Mol Sci. 2023 Feb 3;24(3):3000. doi: 10.3390/ijms24033000; Viruses. 2023 Jan 5;15(1):168. doi: 10.3390/v15010168; Viruses. 2022 Jan 14;14(1):145. doi: 10.3390/v14010145; World J Psychiatry. 2021 Nov 19;11(11):1075. doi: 10.5498/wjp.v11.i11.1075.
Type 1 diabetes: Diabetes Obes Metab. 2020 Jul;22(7):1111. doi: 10.1111/dom.14010 ; Diabetes Obes Metab. 2018 Sep;20(9):2075. doi: 10.1111/dom.13357; Curr Diab Rep. 2019 Nov 21;19(12):141. doi: 10.1007/s11892-019-1256-9.
2. In the “10. HERVs in Nervous System Cancers - The Wicked Side of HERV-W”, the author indicated that “ERVW-1 was demonstrated to stimulate the expression of SK3 and TRPC3”. In fact, ERVW-1 stimulate not only the expression of SK3 and TRPC3, but also Sk2, resulting in an increased potassium ion (K+) current.
See the reference: Virol Sin. 2022 Aug 22:S1995-820X(22)00144-4. doi: 10.1016/j.virs.2022.08.005. PMID: 36007838.
Reviewer 2 Report
This review comprehensively covers the literature relating HERV expression in cancer, classified in 16 human cancers. Summary illustrations should allow readers to follow the complex relationships between HERV expression and cell pathways more easily. Despite some in vitro data that support cause-effect relationship between HERV expression and cancer, I recommend a more conservative position, suggesting the removal of “the Development of” in the title, so that it reads: “A Comprehensive Review of the Relationship of Human Endogenous Retroviruses and Human Cancers”.
Secondly, the readers need justification of the “16 human cancer classification” selected by the authors.
Thirdly, the authors introduce the aims and methods in several sentences (see below few sentences pasted from lines 75 to 93).
Line 75 “we conducted a review on the carcinogenic impacts of HERVs by cancer type.”
Line 89 “we set out to conduct a review of the molecular pathways and mechanisms affected by HERVs across the spectrum of cancer types”
“Therefore, we compared the findings from patients with cancer and cancer cell lines (population) with observations from healthy individuals (comparators).”
“to provide a comprehensive overview of the HERV influences for the 16 human cancer classifications”
I would recommend shortening the presentation of aims and adding a Methods section to describe in detail how were the aims achieved. Is this a narrative rather than systematic review
Lines 1107 and following: the molecular taget of temelimab is MSRV. It should not be confused with syncytin (product or ERVW-1). Please see:
Curtin, F.; Lang, A.B.; Perron, H.; Laumonier, M.; Vidal, V.; Porchet, H.C.; Hartung, H.-P. GNbAC1, a humanized monoclonal antibody against the envelope protein of multiple sclerosis-associated endogenous retrovirus: A first-in-humans randomized clinical study. Clin. Ther. 2012, 34, 2268–2278
Minor details:
Table 1 seems to duplicate some annotations. Classifications “Geneλγπ gene consists of multiple viral genes (see above) geneλθ gene is a viral-human (θ) gene fusion (see above) HERV:geneλθ” included in the table were not found in the text.
Lines 338, 431, 472 and 971: “see Error! Reference source not found”
Spacing format in lines 658 and 659.
